# SemDeDup: Data-efficient learning at web-scale through semantic deduplication

## Abstract

Progress in machine learning has been driven in large part by massive increases in data. However, large web-scale datasets such as LAION are largely uncurated beyond searches for exact duplicates, potentially leaving much redundancy. Here, we introduce SemDeDup, a method which leverages embeddings from pre-trained models to identify and remove "semantic duplicates": data pairs which are semantically similar, but not exactly identical. Removing semantic duplicates preserves performance and speeds up learning. Analyzing a subset of LAION, we show that SemDeDup can remove 50% of the data with minimal performance loss, effectively halving training time. Moreover, performance increases out of distribution. Also, analyzing language models trained on C4, a partially curated dataset, we show that SemDeDup improves over prior approaches while providing efficiency gains. SemDeDup provides an example of how simple ways of leveraging quality embeddings can be used to make models learn faster with less data.

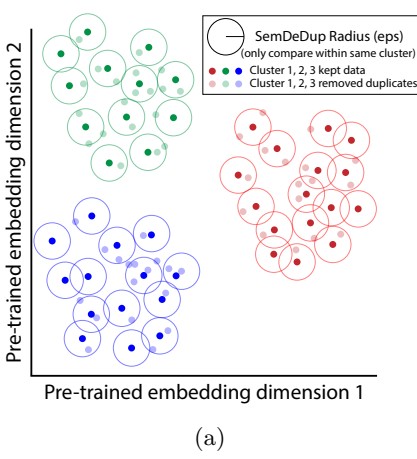

(a)

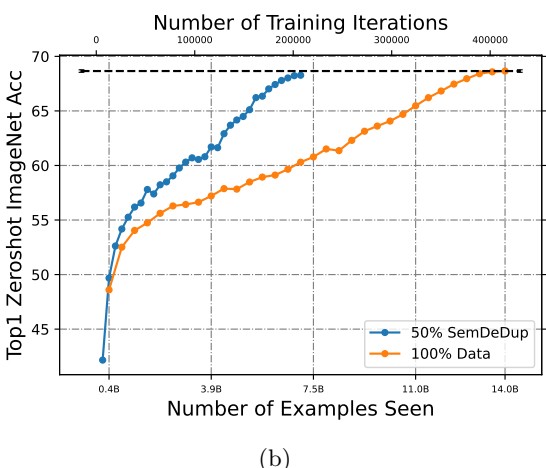

(b)

Figure 1: **Data efficiency from semantic deduplication (SemDeDup)** (a): A schematic of the SemDeDup algorithm which efficiently removes semantic duplicates from web-scale data. (b): When SemDeDup removes 50% of the LAION-440M dataset, training on this semantically *nonredundant* subset achieves almost the *same* performance as training on the *entire* 440M dataset. Also, training speed is *twice* as fast and completes in *half* the time.

## 1 Introduction

A primary driver of recent success in machine learning has been the rise of self-supervised learning (SSL) scaled to ever larger models and unlabelled datasets (Hestness et al., 2017; Kaplan et al., 2020; Henighan et al., 2020; Rosenfeld et al., 2020; Gordon et al., 2021; Hernandez et al., 2021; Zhai et al., 2021; Hoffmann et al., 2022). In particular, modern large datasets are often derived at global web-scale and are generally

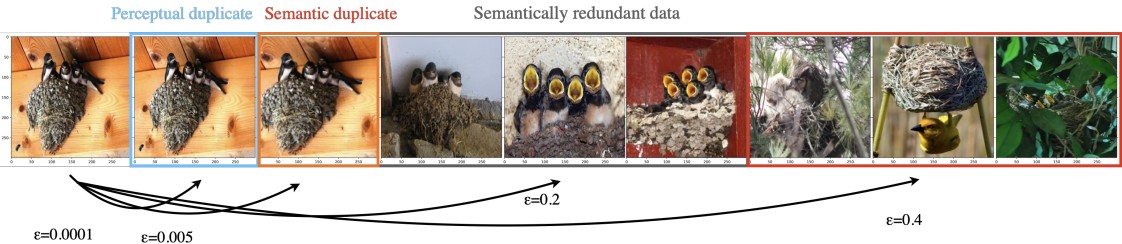

Figure 2: **Mapping cosine similarity to perceptual and semantic similarity.** We visualize pairs of images with cosine similarity $1 - \epsilon$ in the CLIP image encoder embedding space. The left most image is a random seed image from LAION, while the remaining images are sorted by their dissimilarity $\epsilon$ to the seed image. Roughly, as $\epsilon$ increases from left to right, we move from perceptual to semantic duplicates, while at large values of $\epsilon$ we see semantically redundant pairs. Note the red labelled "semantic duplicate" is a view of the original left-most seed image from a slightly different perspective. See more examples in Figure A13.

unfiltered, with the exception of NSFW filters. One such public dataset is LAION (Schuhmann et al., 2022b), a multi-modal dataset of 5 billion image/text pairs. Multi-modal models such as CLIP (Radford et al., 2021b) are trained for many epochs on these large datasets achieving impressive performance but at the cost of extremely long training durations.

The critical role of large datasets has led to increasing interest in scaling laws which enable us to predict how a model's performance will change given more data and/or parameters, leading to the observation that test error generally scales as a power law with respect to data quantity (Kaplan et al., 2020). Power law scaling, however, is unsustainable as diminishing marginal returns are quickly hit such that ever increasing amounts of data are required to achieve ever diminishing improvements in performance. Notably, many of these models appear never to converge, as test performance continues to increase even after 10s of passes through these massive datasets (Ilharco et al., 2021; Aghajanyan et al., 2023). This result suggests that our best models are underfitting, likely as a result of spending an increasing fraction of learning time focusing on redundant data.

Improving data efficiency would therefore be quite impactful, either by enabling models to achieve the same performance much faster, or by enabling models to achieve better performance given the same computational budget. These observations have inspired recent work which suggests that by pruning training data according to an intelligent criterion, power law scaling with respect to data can be beaten and, given an optimal data ranking metric, exponential scaling might in principle be achieved (Sorscher et al., 2022). Recent explorations of this direction have shown promising results, with some works able to reduce data size by almost 5-fold with minimal performance loss (Radenovic et al., 2023).

However, optimal approaches to select data remain poorly understood. Such approaches might focus on one of several different classes of examples to be removed, roughly ordered by the complexity of their discovery:

1. **Perceptual duplicates**: We loosely define such data pairs to be perceptually identical to a typical human observer. The most straightforward version would be exact duplicates at the pixel or token level that could easily be found via exact duplicate detection in input space. However, such approaches might miss pairs of images with human imperceptible pixel level distortions. Most widely-used datasets have some exact duplicate filter already applied, though perceptual duplicates with slight pixel-level differences may pass through such filters.

2. **Semantic duplicates**: these are examples which contain largely identical information content, but remain perceptually distinct. For example, a pair of image views which are derived from the same image, but feature different margins, aspect ratios, color distributions, etc. could be considered semantic duplicates. A pair of sentences with the same structure but some words exchanged for synonyms would also be considered a semantic duplicate. Such pairs would rarely, if ever, be detected by exact duplicate filters as they would be far apart in pixel/token space.

3. **Semantically redundant data**: in contrast to semantic duplicates, semantically redundant data are not derived from the same underlying objects and would be clearly distinguishable to a human. However, the information contained in such examples may still contain substantial overlap. For example, consider the case of two different images of two different golden retrievers in two different parks. These images are neither perceptually nor semantically identical as the content of the images differs. However, the information contained in them is quite similar, leading us to think of such pairs as semantically redundant. Each additional semantically redundant data point will provide less and less new information, eventually converging to near-zero information gained from additional such data. Methods such as SSL Prototypes (Sorscher et al., 2022) and memorization (Feldman & Zhang, 2020) search for semantically *non-redundant* data subsets to train on.

4. **Misleading data**: these are data which rather than providing zero information (as in the previous categories) provide negative or *harmful* signal, in the sense that removing these data actually improves performance, rather than having a neutral effect. While such data are easy to conceive of in supervised learning (i.e. mislabeled examples), it is much less clear what such examples may be in the context of self-supervised learning.

In this work, we focus on the category of semantic duplicates: data which are semantically highly similar but which would be difficult to discover using simple deduplication approaches. These data points are challenging to identify because distance measures in input space are unlikely to uncover semantic duplicates. To overcome this limitation, we leverage pre-trained foundation models to compare data similarity in the learned embedding space rather than in input space. Comparing every data point to every other data point, however, is intractable, especially for web-scale datasets containing billions of examples. To make this computation possible, we use the clustering approach described in (Sorscher et al., 2022) to segment the embedding space, allowing us to only search for duplicate pairs within a cluster. Using this approach, we make the following contributions:

- We propose SemDeDup [1] (Fig. 1, a), a simple, yet effective and computationally tractable way to identify semantic duplicates. Using this approach, we show that large web-scale datasets such as LAION contain large numbers of semantic duplicates, with 50% of examples containing at least one semantic duplicate that can be removed without losing performance.

- Large fractions of semantic duplicates can be removed with little-to-no performance impact, greatly increasing training efficiency. We reduced the size of our LAION training set by 50% with minimal performance loss, and improved learning speed, achieving nearly the same performance 2x faster (Fig. 1, b), and moreover *improved* performance out-of-distribution.

- We apply SemDeDup to C4, a large text corpus, beating prior SoTA deduplication while providing efficiency gains of 15%, sometimes even improving performance.

Overall, our results demonstrate a simple yet surprisingly effective approach to reduce the cost of training through the removal of semantic duplicates which is likely applicable to all web-derived datasets and may help to democratize the training of large-scale foundation models by improving data and compute efficiency.

## 2 Related Work

Much of the work in language and vision on deduplication has focused on the removal of exact duplicates. For example, (Liao, 2022) removed duplicates between the YFCC15M dataset (Thomee et al., 2016) and the ImageNet validation set to prevent train-test leakage. The C4 text corpus - used for training T5 (Raffel et al., 2019) - has been deduplicated by discarding repeated occurrences of any three-sentence spans. (Lee et al., 2021a) showed that it's possible to further deduplicate this dataset without loss of performance by computing approximate n-gram overlap between documents using the MinHash technique (Broder, 1997).

---

[1]We call the method SemDeDup (Semantic De-Duplication), but that includes also identifying exact and near-duplicates, as they exhibit a higher level of similarity. For more detailed definitions, please refer to Section 1.

(Rae et al., 2021) also applied MinHash based deduplication to curate training data for the Gopher model and demonstrated that training on the deduplicated dataset can result in lower perplexity across various validation sets. (Kandpal et al., 2022) found that deduplication prevents memorization in LLMs and thus mitigates privacy concerns. More recent works use forms of model-based feature extraction to improve the robustness of the similarity metric used for deduplication. (Silcock et al., 2022) created a supervised dataset for detecting duplicate news articles and trained models to predict those labels. In the domain of computer vision, (Choi et al., 2022) improves on SSL techniques by removing near-duplicates in some high dimensional feature space they learn.

Beyond deduplication, a host of classical machine learning approaches seek to achieve data efficiency by finding *coresets*, defined as small subsets of the training data that can be used to train a machine learning algorithm to the same test accuracy achievable when training on the entire training data (see e.g. (Guo et al., 2022; Phillips, 2016) for reviews). However, many coreset algorithms are computationally prohibitive and therefore are difficult to scale to web-scale data. In contrast to many traditional coreset algorithms, we develop an exceedingly simple and tractable algorithm that achieves both computational and data efficiency.

Recent approaches to achieve data efficiency in deep learning have operated in a supervised setting by defining and finding "hard" examples not easily learned by partially or fully trained (ensembles of) models (Toneva et al., 2019; Paul et al., 2021; Chitta et al., 2021; Feldman & Zhang, 2020; Meding et al., 2022). (Sorscher et al., 2022) proposed using clustering in the embedding space of a pre-trained foundation model to rank examples based on their usefulness for training. Their method aims to remove prototypical examples within each cluster by ranking examples without considering their semantic similarity to each other, but rather by comparing each to the cluster centroid and removing those points which are closest to the centroid. As a result, pairs of duplicates are assigned similar ranks, and they are either kept or removed together. In this work, we also utilize clustering, but we instead aim to remove semantic duplicates by focusing on nearby points regardless of their location. Furthermore, we move from relatively small, highly curated ImageNet scale to highly uncurated, web-scale data. Our analysis, at this new large and uncurated scale, reveals a possibly fundamental role for semantic deduplication as an important initial step in data-pruning for self-supervised learning that was not considered in prior data curation works.

**Methods for de-duplication.** Several techniques have been proposed for image de-duplication. One approach involves hashing the data in the input space. For instance, (Lee et al., 2021b) utilizes a minhash algorithm to hash text documents and identify near-duplicates. (Li et al., 2021) applies Perceptual Hashing (phash), Average Hashing (ahash), Difference Hashing (dhash), and Perceptual Hashing (phash) to hash image datasets.
While hashing methods offer fast execution, they have certain limitations. For example, they require parameter settings and many of them are weak in capturing semantic features of the data. Moreover, different hashing functions are typically needed to be designed for different modalities (e.g., image, text, audio, etc.).
Another category of techniques involves utilizing embeddings from pre-trained models to represent the data. The idea of working in the embedding space has been utilized in many works for coreset selection (Sinha et al., 2020), (Sener & Savarese, 2017) (Agarwal et al., 2020). Unlike hash codes, embeddings are typically larger in size but offer a richer representation of the semantic information of the data. (Webster et al., 2023), and Jain et al. (2019) apply deduplication in the embedding space on different scales but without providing strong study of training on deduplicated data. The *fastdedup* library also offers techniques for deduplicating large image datasets, but there is a lack of studies on training using deduplicated datasets.

## 3 SemDeDup

**Defining and identifying semantic duplicates**   While identifying perceptual duplicates can be easily done in input space, identifying semantic duplicates is more difficult as they may be distant in either pixel or token space. To identify these pairs, we leverage the embedding space of a large pre-trained foundation model to provide a more semantically meaningful distance metric. To detect and remove semantically similar images, we use the following semantic de-duplication (SemDeDup) algorithm (Fig. 1, a). First, we embed each data point using a foundation model (CLIP (Ilharco et al., 2021; Radford et al., 2021a) for images and OPT (Zhang et al., 2022) for language). We then cluster the embeddings into $k$ clusters via k-means. Below,

we choose $k = 50,000$ clusters in CLIP image encoder embeddings and $k = 11,000$ clusters in OPT-language model embeddings. Within each cluster, we compute all pairwise cosine similarities and set a threshold cosine similarity above which data pairs are considered semantic duplicates. Finally, from each group of semantic duplicates within a cluster, we keep one example only. To choose which example to keep exactly, we follow the hypothesis in (Sorscher et al., 2022) that examples with the lowest cosine similarity to the cluster centroid are better for training, so we keep the closest example to the centroid from each group of duplicates and remove the rest. We show later in Section (6) that SemDeDup is robust to how exactly we choose the example.

We note that to determine duplicates, for LAION dataset, this method considers only the images and ignores the captions. A simplified pseudo code for SemDeDup is shown in Algorithm A10 in the appendix. We provide more details about the method in addition to experiments on choosing the value of $k$ in Section 6.

**Utilizing pre-trained foundation models.** Our method makes use of pre-trained foundation models to embed data examples. Considering that there are many of these ready-to-use pre-trained models available to the public, we can use embeddings from these models to guide curation of other datasets. Pre-trained models like Vision Transformers (Dosovitskiy et al., 2020) for vision tasks, OPT (Zhang et al., 2022) for natural language and CLIP (Radford et al., 2021a) for vision-language data have been used widely. In this work, we utilize pre-trained CLIP and OPT models for deduplication. In addition, in Section 6, we show that one can effectively use an on-the-shelf model pre-trained on one dataset to prune another dataset resulting in a considerable training cost saving.

**Clustering to reduce computation** Naively searching for duplicates has a time complexity of $O(n^2)$ where $n$ is the number of data points. Comparing all examples in a dataset to each others can find all duplicates, but this approach is impractical for large web-scale data. For example, the LAION-440M dataset would require $\approx 1.9\text{x}10^{17}$ similarity computations. Our primary objective is therefore to reduce the computational cost of these comparisons. The k-means clustering step in SemDeDup reduces the complexity of searching for duplicates substantially from $O(n^2)$ to $O(\frac{n^2}{k})$ assuming approximately uniform cluster size[2]. This means we only require $\approx 3.9\text{x}10^{12}$ intra-cluster comparisons instead of $\approx 1.9\text{x}10^{17}$ across all pairs, a 5-order of magnitude improvement.

Considering that k-means clustering has a time complexity of $O(k.n)$[3] (See Appendix A.1 for more details). The time complexity of SemDeDup is upper bounded by the clustering step complexity ($O(kn)$) or by the search step complexity ($O(\frac{n^2}{k})$). The search step dominates when $\frac{n^2}{k} > kn$ or simply when $k < \sqrt{n}$ and the complexity of SemDeDup in this case is $O(\frac{n^2}{k})$. While the clustering step dominates when $k \geq \sqrt{n}$ resulting in a time complexity of $O(k.n)$. As long as $1 < k < n$ the time complexity of SemDeDup ($O(\frac{n^2}{k})$ or $O(kn)$) stays below $O(n^2)$. In numerical terms, this corresponds to corresponds to $\approx 3.9\text{x}10^{12}$ or $\approx 2.2\text{x}10^{13}$ pairwise comparisons respectively, for LAION-440M dataset when $k = 50,000$, which is 5 or 4 orders of magnitude faster than conducting $n^2$ ($\approx 1.9\text{x}10^{17}$) comparisons.

## 4 SemDeDup on LAION

If we consider pairs of data points to be semantic duplicates when their cosine similarity is at least $1 - \epsilon$, then $\epsilon$ can be thought of as a deduplication dissimilarity threshold, with increasing $\epsilon$ reflecting an increasingly coarser notion of semantic equality. We expect that low thresholds of $\epsilon$ will find semantic duplicates, while higher thresholds will allow semantically redundant data pairs as well.

To evaluate SemDeDup's ability to discover semantic redundancy in multi-modal data, we train CLIP models on the LAION dataset (Section 4.1). We first show that LAION contains extreme amounts of semantic redundancy (Section 4.2) and provide examples of the semantic duplicates discovered by SemDeDup (Section 4.3). Most critically, we demonstrate that removing the semantic duplicates discovered by SemDeDup has minimal to no impact on converged performance and increases learning speed (Section 4.4).

---

[2]Note that our choice of $k$ depends on $n$, it is not a constant in the context of this complexity analysis.

[3]The naive k-means clustering has a time complexity of $O(i.k.n)$ ($i$ is the number of interations), but instead we use an efficent k-means clustering implementation from $faiss$. See Appendix A.1 for more details about time complexity of k-means clustering.

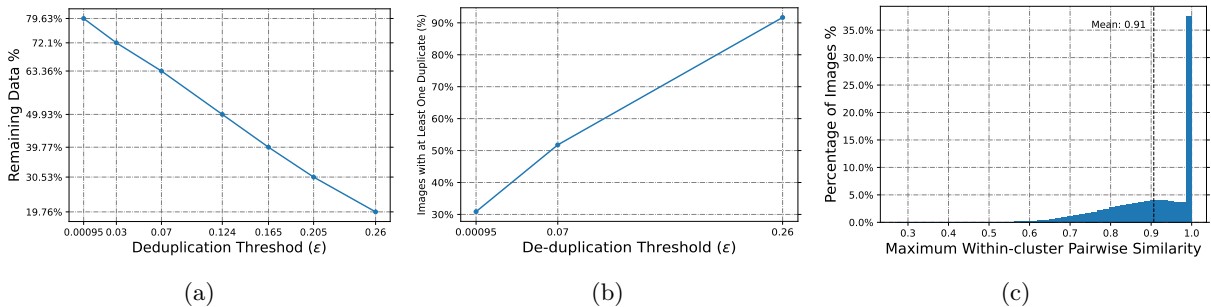

Figure 3: **Extreme semantic redundancy in LAION-440M.** (a) Fraction of data remaining as a function of deduplication threshold $\epsilon$ for LAION-440M. (b) Percentage of images in LAION-440M with at least one semantic duplicate as a function of $\epsilon$. (c) Histogram of the number of within-cluster image pairs in LAION-440M at a given cosine similarity.

### 4.1 Datasets and Training

**The LAION dataset.** To train large-scale multi-modal models, we used the LAION dataset (Schuhmann et al., 2022a), an open multi-modal dataset containing up to 5 billion image-text pairs scraped from the web. LAION data were filtered using a pre-trained CLIP model to only retain image-text pairs with an embedding similarity greater than 0.28. Image-text pairs containing very short captions or small images were also removed. A simple de-duplication method based on the image url was also performed.

The majority of our experiments were performed on the LAION-440M filtered subset of LAION-2B introduced by (Radenovic et al., 2023). This dataset was filtered using a Complexity, Action, and Text (CAT) filtering according to three criteria: (1) high enough caption **complexity**; (2) the caption must contain an **action**; (3) any **text** present in the image cannot substantially overlap with the caption.

To ensure this CAT filtered LAION-440M subset did not impact our results, we also performed experiments on unfiltered data derived from LAION. Much of the original LAION-400M subset (Schuhmann et al., 2021) is no longer available due to broken urls, so we used a reduced version of the LAION-400M subset containing the 233 million data points we were able to collect, which we call LAION-233M.

**CLIP training.** For CLIP training on LAION, we use the OpenCLIP implementation (Ilharco et al., 2021). We use CLIP-ViT-Base-16 in all our experiments. The model has Vision Transformer Base (ViT-B-16) (Dosovitskiy et al., 2020) as an image encoder and Text Transformer (Vaswani et al., 2017) as a text encoder. We train all models with a global batch size of 33,792 (33k for simplicity) image-caption pairs and fix the number of training epochs to 32 regardless of the dataset size. This results in training for a fewer number of iterations when training on deduplicated data, thereby achieving efficiency gains. For example, training on the LIAON-440M dataset for 32 epochs corresponds to 14B examples seen during training, and training on 50% of it for 32 epochs corresponds to 7B (14 x 0.5) examples seen.
We train with AdamW (Loshchilov & Hutter, 2017) and cosine learning rate schedule with warmup. The same peak learning rate of $5\times10^{-4}$ is used for all models. Table A6 shows training parameters we use for CLIP. In addition to fixing the batch size for all models, we also fix the number of GPUs used for training, specifically we use 176 A100 GUPs.

**CLIP Evaluation** For CLIP evaluation, we conduct zeroshot evaluation on 30 distinct datasets, of which six are designated for assessing out-of-distribution (OOD) robustness. Tables A7 and A8 in the Appendix list all the datasets we use for evaluation.

### 4.2 Extreme semantic redundancy at web-scale

How many semantically redundant pairs are there in LAION? Remarkably, we find that even tiny thresholds $\epsilon$ lead SemDeDup to remove large fractions of data in LAION-440M (Fig. 3a), showing that LAION-440M

contains large quantities of semantic duplicates. Surprisingly, 30% of images in LAION-440M have a semantic duplicate at the highly stringent distance threshold of $\epsilon = 0.00095$, while 50% have a duplicate at the tight threshold of $\epsilon = 0.124$ (Fig. 3b). Moreover, a histogram of pairwise cosine similarity in LAION-440M (Fig. 3c) reveals a high density of pairs at high cosine similarity, including a large contribution at 1, reflecting highly similar semantic duplicates. These results demonstrate that LAION-440M contains large amounts of semantic redundancy.

### 4.3 What do semantic duplicates look like?

What leads to semantic duplicates? In Fig. 2, we show examples of semantic duplicates found at different thresholds $\epsilon$. At extremely low values of $\epsilon$ we find perceptual duplicates, and at slightly higher values of $\epsilon$, we find semantic duplicates, which are the same image but with distortions which evade exact de-duplication approaches such as different margins, crops, aspect ratios, and color filters, or slightly different peripheral details. Fig. A14, and A15 show examples of clusters that are semantically deduplicated at increasing levels of $\epsilon$, clearly indicating more semantic diversity in deduplicated clusters as $\epsilon$ increases.

Many semantic duplicates are of products which may have been displayed on multiple e-commerce websites, each with a slightly different style. As a result, semantic duplicates often contain different, but highly similar captions. While most clusters contained 20-40% duplicates, there are several remarkable outliers in redundancy in LAION-440M (Fig. A12), including one cluster containing $\approx 307,000$ copies of the European Union flag and another with $\approx 318,000$ copies of an icon of "Image not found".

At higher levels of $\epsilon$ in Fig. 2, and A13, we find fewer semantic duplicates, which are generally derived from the same source image, and more pairs which exhibit semantic redundancy instead, in which the same concept is present, but not derived from the same image source. For example, semantically redundant pairs may contain different images of similar objects or scenes.

### 4.4 Training on semantically deduplicated data improves efficiency

If SemDeDup is effective at finding semantic duplicates, we should be able to remove these duplicates with a minimal performance impact. To test this, we train CLIP models on subsets of LAION-440M deduplicated at different thresholds $\epsilon$, corresponding to smaller fractions of data as $\epsilon$ rises. For SemDeDup, we use embeddings from a CLIP model pretrained on LAION-440M. We show later in Section 6.2 that SemDeDup is robust to the choice of the pretrained model.

In Fig. 4 (a), we plot the top-1 zero-shot accuracy of our CLIP models on ImageNet-1k. Encouragingly, we found that SemDeDup can remove up to 37% of LAION-440M with no performance drop, and 50% with minimal performance drop ($< 0.5\%$). In contrast, randomly removing data results in much larger drops. In Fig. 4 (b), we show the average zero-shot performance across 24 tasks, finding that on average, performance increased on de-duplicated data. See Table A7 for detailed performance on all 24 tasks at 6 deduplication thresholds as well as 1 baseline and 4 random controls. See Fig. A8 for performance on 24 individual tasks.

We also evaluated out-of-distribution robustness on 6 datasets commonly used for this task: ImageNet-A, ImageNet-O (Hendrycks et al., 2021b), Imagenet-R (Hendrycks et al., 2021a), Imagenet-sketch (Wang et al., 2019), ImageNetV2 (Recht et al., 2019), and ObjectNet (Barbu et al., 2019). We again found that SemDeDup *increased* average performance over baseline when *removing* 37% of the data, and matched performance when 50% was removed as shown in Fig. 5 (a). See Table A8 for detailed performance on 6 OOD tasks at 6 deduplication thresholds as well as 1 baseline and 4 random controls. We also note that SemDeDup outperforms random pruning on all individual out-of-distribution robustness datasets for all fractions of dataset kept.

Fig A1 shows SemDeDup performance across 24 zero-shot tasks when removing 37% of the data, relative to a CLIP baseline trained on all the data. On about 15 out of 24 tasks, performance actually *improves* after *removing* pre-training data, whereas on all but about 3 of the remaining tasks performance is not substantially reduced. Our observation that SemDeDup can improve performance in many cases is consistent with prior work which has found that removing duplicates may improve performance by discouraging memorization (Lee et al., 2021b).

We emphasize that SemDeDup achieves these results on LAION-440M, an already highly curated dataset derived from LAION-2B which was found to have similar performance despite the almost five-fold reduction in data (Radenovic et al., 2023). However, to ensure that this curated subset did not bias our results, we also evaluated on LAION-233M, an uncurated subset of LAION-2B, finding qualitatively similar results (Fig. A10).

Because SemDeDup reduces the number of training points, it enables substantially faster training. In Fig. 5 (b), we plot the top-1 zero-shot accuracy on ImageNet-1k as a function of the number of iterations for different deduplication thresholds $\epsilon$. Notably, models trained on deduplicated data reach convergence in substantially fewer iterations.

### 4.5 Why do models trained on uncurated data exhibit slower learning?

We posit that successive learning iterations involving semantic duplicates yield redundant information, thereby wasting valuable computation on data points that are highly similar to those the model has already seen. By removing these semantic duplicates, we increase the fraction of data points which provide a marginal information gain to the model, thereby increasing learning speed (Sorscher et al., 2022). In addition, duplication makes the dataset more cluster-balanced by removing more examples from large clusters (Fig. A11). It has been shown that training on imbalanced datasets can have negative effects on a model's ability to generalize (Liu et al., 2021), (Wang et al., 2023). Our empirical results in Fig. (4) and (5) suggests that using deduplicated data can outperform training on the whole uncurated data when the deduplicated dataset size is large enough (63% for LAION), especially on out-of-distribution detection tasks (Fig. 4 and 5). All models are trained using the same number of GPUs and batch size.

### 4.6 Training on deduplicated data for more iterations improves performance

Training on deduplicated data comes with the advantage that we train for fewer iterations when training for the same number of epochs. For example, training on 50% of LAION-440M for the same number of epochs as the baseline model (100% of the data) means that we train for only 50% of the number of training iterations. We find that we can achieve a good trade-off between performance and training speed when training on deduplicated data. We show that training on deduplicated LAION-440M for more iterations improves the accuracy while still being below the number of iterations we train the baseline model for. In Table A1, we show results for different CLIP models, trained on 50% of LAION-440M, for a different number of training iterations. We see that by continuing training the model until we reach 75% of the iterations relative to the baseline model, we outperform the baseline model on not only ImageNet, but also on average accuracy over 24 datasets, and on the 6 out-of-distribution datasets.

## 5 SemDeDup on Natural Language

### 5.1 Methods

We train language models on deduplicated versions of the C4 dataset (Raffel et al., 2019). Since pre-training large language models on the entire C4 corpus is beyond our compute budget, we train on subsets of this data whose sizes are compute optimal given model size as per (Hoffmann et al., 2022). We use the OPT model and training configurations (Zhang et al., 2022) to train 125M and 1.3B parameter models (see Table 1 in (Zhang et al., 2022) for full specifications). We use the original number of warmup updates but adjust the learning rate schedule such that all training runs anneal learning rate to 0 by the end of the training — this allows for fair comparisons of model performances across different dataset sizes. For 1.3B model size experiments, we increase the number of warmup updates to 5550 and reduce the peak learning rate to $6x10^{-5}$ to stabilize training.

We evaluate our trained language models on two independent validation sets: the validation text corpora used by OPT (Zhang et al., 2022) (referred to as "opt_valid") and a random sample of the instruction finetuning corpus used to train the OPT-IML family of models (Iyer et al., 2022), composed of verbalized prompts corresponding to a wide range of NLP tasks and their solutions (referred to as "prompts_with_answers").

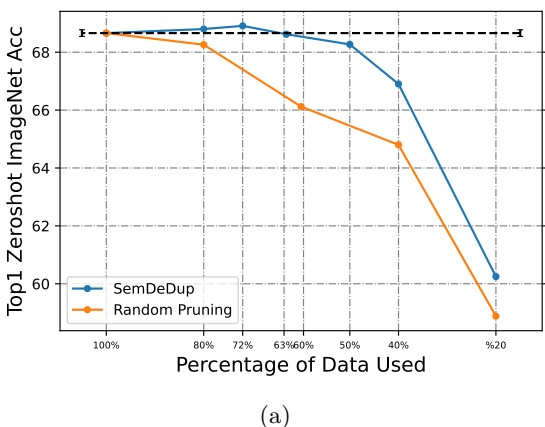

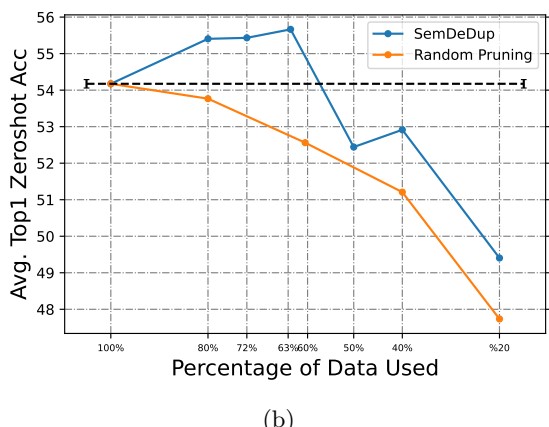

(a)

(b)

Figure 4: **SemDeDup allows better average zero-shot accuracy across 24 tasks with less data and faster pre-training.** (a): Performance of SemDeDup (blue) and random pruning (orange) for different amounts of retained data. Down to using only 50% of LAION-440M for pre-training CLIP, we are able to match the zero-shot ImageNet accuracy of the baseline model trained on 100% of the data (black dashed line) with a small drop of 0.38% only, while we outperform the baseline model with only 63% of data. (b): Average zero-shot performance for CLIP measured on 24 datasets. Average performance *improves* across 24 tasks down to 63% of the pre-training data, yielding *better* performance with almost $1.6\times$ *faster* pre-training. The black dashed line corresponds to the average performance over 4 different traning seeds for a baseline model trained on 100% of the data. The error bars indicate the standard deviation (std) of performance.

To perform SemDeDup, we pass documents through the open-sourced pre-trained 125M OPT model (Zhang et al., 2022) and save the last layer embedding for the last token in the document. We then apply the same method described in Section 3 with $\mathcal{K} = 11000$ to cluster these embeddings. We compare to random pruning and the NearDup method described in (Lee et al., 2021b). Note that the deduplication threshold values associated with different fractions of data remaining change compared to LAION-440M, as seen in Fig. A21.

## 5.2   Results on Language Modeling

In Fig. 6, we show the performance of SemDeDup versus random pruning. We observe that SemDeDup significantly outperforms random pruning as measured by perplexity on prompts_with_answers and average opt_valid performance. For a breakdown of performance on individual validation sets in opt_valid, see Fig. A24 where we observe that SemDeDup beats random pruning on every single validation set in opt_valid.

Training on less data for one epoch naturally causes performance to decrease. Thus, we also explore whether continuing to train on the same smaller pruned datasets for more epochs will match the performance of a baseline model trained on a larger dataset. In Fig. A4, we train on datasets pruned with SemDeDup, but perform the same number of total training steps as the baseline model on the larger dataset (which was trained for 1 epoch). This causes the model to do multiple epochs over the pruned dataset. We observe that by training for multiple epochs over significantly pruned datasets we can reach the performance of a single-epoch run on the full dataset using 10-15% less compute. This is similar to the finding in Section 4.4. Notably, this efficiency gain is larger at higher pruning percentages, indicating that more aggressive pruning can yield more efficiency gains. This trend generally holds across the individual validation sets in opt_valid (see Fig. A25).

On the C4 validation set, we observe that SemDeDup still outperforms random pruning in Fig. A22. In Table A16 we compare SemDeDup to the NearDup baseline from (Lee et al., 2021a). We observe that NearDup and SemDeDup have comparable performance as is expected, because with 4% pruning there is very little change to the underlying dataset.

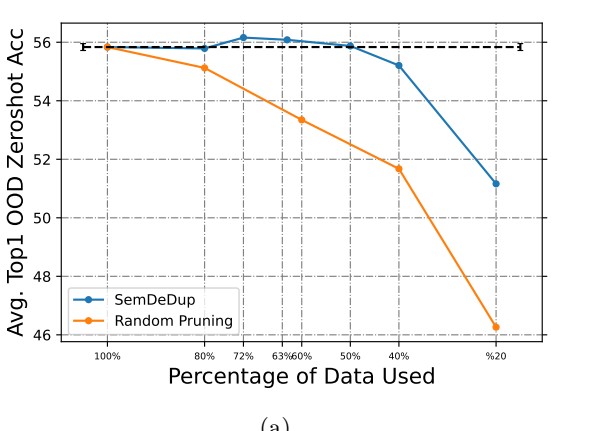
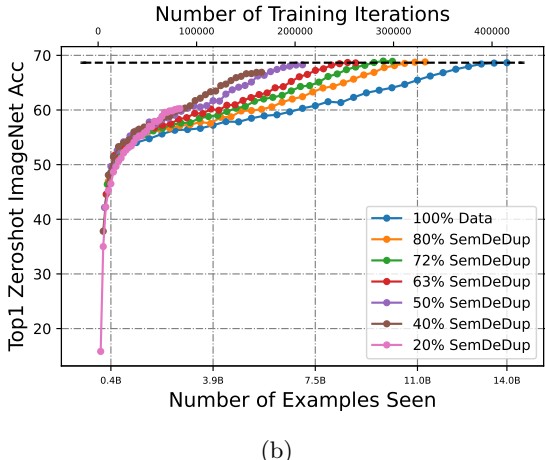

(a)                                                                  (b)

Figure 5: **SemDeDup allows better average performance across 6 ImageNet OOD tasks with less data and faster pre-training. (a)** zeroshot validation accuracy averaged over 6 ImageNet-1k OOD tasks for CLIP models pre-trained on deduplicated LAION data with different thresholds $\epsilon$. We outperform the baseline model with only 63% of pre-training data from LAION-440M. **(b)** We track zeroshot ImageNet-1K performance as a function of LAION-440M pre-training iterations at different deduplication thresholds. The models trained on smaller deduplicated datasets actually learn faster, thereby allowing them to converge to almost baseline performance (black dashed line) in far fewer iterations. The black dashed line corresponds to the average performance over 4 different traning seeds for a baseline model trained on 100% of the data. The error bars indicate the standard deviation (std) of performance. All models are trained using the same number of GPUs and batch size.

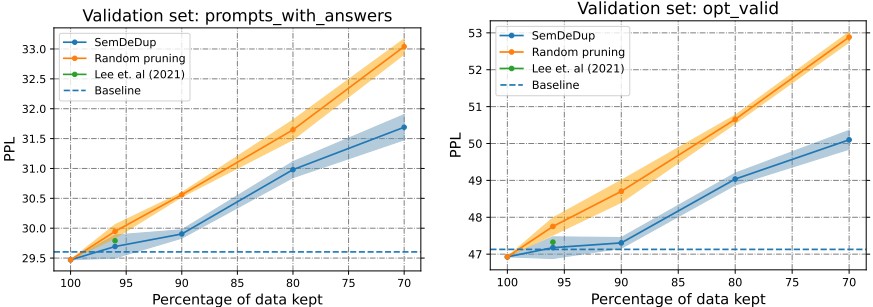

Figure 6: **SemDeDup applied to C4**. The x-axis corresponds to different percents of data kept, and the y-axis represents the perplexity on validation sets described in Section 5.1 (lower is better). Each point is a separate 125M model trained on one-pass of its respective pruned dataset (mean and standard deviation across 3 random training seeds). The green point represents a 125M model trained on a version of C4 deduplicated via the NearDup method (Lee et al., 2021a). Note that NearDup (the single green point) keeps 96.1% of the data. SemDeDup can match this baseline performance while keeping only 80% of the data (see Table A17 for numerical comparison).

## 6 Analysis of hyperparameter choices

### 6.1 Number of k-means clusters for SemDeDup

Here we study the impact of changing the number of clusters $k$ in the k-means clustering step in SemDeDup described in Section 3. In all our experiments in the main paper, we set $k = 50,000$ for the LAION dataset and $k = 11,000$ for the C4 dataset. To study the impact of $k$ on the performance, we deduplicate LAION-440M

Table 1: Performance of CLIP when keeping 40% of LAION-440M as a function of the number of k-means clusters $k$ used for SemDeDup. SemDeDup is robust to the choice of $k$ and the impact on the zeroshot accuracy on ImageNet is small with slight performance improvement as we increase $k$.

| Metric / Num. of Clusters | 70K Clusters | 50K Clusters | 10K Clusters |
|---|---|---|---|
| Top1 zeroshot IN Acc. | 67.11 | 66.90 | 66.56 |
| Top5 zeroshot IN Acc. | 90.96 | 90.74 | 91.04 |

Table 2: The impact of the foundation model used for extracting embeddings. Interestingly, it is enough to use a DINO ViT-Base model pre-trained on ImageNet only to embed LAION-440M.

| Metric / Model Used for Extracting Embeddings | CLIP Pre-trained on LAION-440M | OpenAI CLIP Pre-trained on WIT-400M dataset | DINO Pre-trained on ImageNet (1.2M) |
|---|---|---|---|
| Top1 zeroshot IN Acc. After Training on DeDup. Data | 66.90 | 66.96 | 66.77 |
| Top5 zeroshot IN Acc. After Training on DeDup. Data | 90.74 | 90.80 | 90.86 |

using different values for $k$ and train different CLIP models on the deduplicated data. We compare three values for $k$ (70,000, 50,000, and 10,000) when deduplicating LAION-440M to 40% of its size. As we see in Table 1 the exact choice of $k$ has a very small impact on performance as measured by the zeroshot accuracy on ImageNet with a small improvement in the top1 accuracy as $k$ increases.

The key intuition is that the choice of $k$ implements a tradeoff in the probability of recovering all semantic duplicates of any data point, and the computational complexity of doing so. For example, assuming k-means finds equal cluster sizes, each data point will lie in a cluster of size $N/k$, and we are only searching for $\epsilon$-nearest neighbors (with cosine similarity $> 1 - \epsilon$) within each cluster. As $k$ decreases, cluster size $N/k$ increases, and the error probability of substantially many $\epsilon$ nearest neighbors of a data point lying outside it's own cluster decreases, while the computational complexity of searching for all nearest neighbors within the cluster increases. As long as $k$ is small enough relative to the total dataset size $N$, so that $N/k$ is large enough to contain most nearest neighbors of each data point, the performance of SemDeDup should be robust to the choice of $k$.

## 6.2 Pre-trained models for extracting embeddings

As we describe in Section 3, SemDeDup clusters the embeddings extracted from a pre-trained foundation model and uses them for deduplication. To study the effect of the pre-training dataset of the foundation model on SemDeDup we deduplicate LAION-440M using different models pre-trained on a different dataset than LAION. We test with a self-supervised DINO ViT-Base-16 (Vision Transformer Base) model (Caron et al., 2021) pre-trained on a dataset as small as ImageNet as well as an OpenAI CLIP ViT-Base-16 model (Radford et al., 2021a) pre-trained on a private dataset of 400 million image-caption pairs (WIT-400M). We use the embeddings from these models to deduplicate LAION-440M dataset to 40% of its size. As we see in Table 2, using any of these models for extracting embeddings has a negligible impact on performance.

## 6.3 Different strategies for choosing which semantic duplicates to keep

In Section 3 and Algorithm A10, we describe the steps for deduplication with SemDeDup. From each group of duplicates (the circles in Figure 1), we keep the example with the lowest cosine similarity to the cluster centroid in the embedding space. This is the default setting for all experiments we run unless otherwise mentioned. In Table 3 we study the strategy we follow to choose the example to keep from each group of duplicates. We train three CLIP models on 40% of LAION-440M deduplicated by SemDeDup for 32 epochs. We try three options for choosing the examples we keep 1) keeping examples with *low* similarity to centroids, 2) keeping *random* examples, and 3) keeping examples with *high* similarity to cluster centroids. We obverse that the difference between the three methods in zero-shot accuracy on ImageNet is negligible.

Table 3: Different strategies to choose the example to keep from each group of duplicates.

| Metric / Examples to Keep | Examples with low similarity to centroids | Random examples | Examples with high similarity to centroids |
|---|---|---|---|
| Top1 zeroshot IN Acc. | 66.90 | 66.90 | 66.73 |
| Top5 zeroshot IN Acc. | 90.74 | 90.95 | 90.82 |

**6.4**

# 7 Discussion

We introduced SemDeDup, a simple yet tractable and effective method which leverages pre-trained embeddings to remove semantic duplicates which are highly semantically similar but not identical. Removing semantic duplicates improves learning speed and out-of-distribution performance while providing efficiency gains of up to 50% on the largely uncurated LAION and 15% on the partially curated C4. SemDeDup demonstrates the importance of data quality and the potential of data curation to dramatically improve training efficiency.

Another important application of data duplication methods like SemDeDup which goes beyond outperforming training on the entire dataset, is selecting high-quality subsets from large datasets, which can be beneficial in various scenarios. For instance, when training models that don't require the entire dataset or when academic researchers face computational limitations. In such cases, accessing high-quality subsets from large datasets becomes advantageous. For example, given a large dataset like LAION (that exceeds 5 billion examples), we can select a small subset (e.g. 5% of the dataset) to train a model that requires less compute. The goal here is not to make the 5% dataset superior to the entire dataset for training. Instead, the primary goal is to find a relatively small subset that does not contain exact, near, and semantic duplicates. Hence, we perceive the datasets obtained by deduplicating the LAION-440M dataset to sizes such as 40% or 20% (or any other size) as highly valuable datasets for training diverse self-supervised models.

**Limitations.** While SemDeDup does an effective job of removing semantic duplicates and semantically redundant data points, it is only one way to remove uninformative data points. In particular, this work does not capture many aspects of semantic redundancy, nor does it address removal of bad or misleading data, all of which can likely be exploited to make further reductions to dataset size without sacrificing performance. SemDeDup also requires access to a pre-trained embedding model relevant to the domain of interest, which may pose a problem for entirely novel domains unrelated to the wide array of publicly available pre-trained models. However, for most domains, pre-trained models are readily available, and many such models have been shown to generalize to related domains. We, therefore, expect that this limitation will only apply to a small fraction of the practical use cases for SemDeDup.

In LAION, we identified semantic duplicates based only on image data, but we ignored the caption information. Leveraging this information may lead to the identification of further semantic duplicates.

Our results on C4 showcase the potential of SemDeDup for NLP, but the gains were more modest due to the partially curated nature of C4 which has fewer duplicates than LAION. We also trained small models relative to the best models. It is possible that results may change with scale, though following (Sorscher et al., 2022), it is likely increasing scale would further improve the benefits of data curation.

Overall, the optimal data pruning policy for finding the smallest possible data subset under computational tractability and performance constraints remains, as ever, an extremely difficult open question. However, the remarkable efficacy of SemDeDup, especially given its underlying simplicity and scalability, suggests that the removal of semantic duplicates may well be an important prerequisite for any more sophisticated data pruning algorithm, especially when working with modern, large, highly uncurated, web-scale datasets.

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

# A    Additional Analysis

## A.1    K-means Clustering

We use the $faiss$ library Johnson et al. (2019) for clustering. $faiss$ is a library for efficient clustering on millions of vectors with GPU support. We use Spherical k-means (Dhillon & Modha, 2000) as we found it better for clustering on ImageNet. Spherical k-means normalizes the cluster centroids after every iteration to have a unit length. This requires the data to also be normalized before clustering. In all our experiments, we run 100 clustering iterations for LAION-440M and 20 iterations for C4. We found that centroids do not move after this number of iterations.

**K-means Clustering Time Complexity**    The time complexity of k-means clustering is $O(i.k.n)$, where $i$, $k$, and $n \in \mathbb{Z}^+$ are the number of clustering iterations, number of clusters, and dataset size respectively. For large scale datasets $i.k < n$. In practice, we can use a small subset of the data to compute cluster centroids for large scale datasets. We use $faiss$ library (Johnson et al., 2019) which efficiently computes centroids on a random subset of $m.k$ examples from the dataset where $m$ is choosen so that $m.k < n$. We choose the values of $i$, $m$, and $k$ to be 100, 256, and 50,000 respectively. This approach reduces the number of required comparisons for computing k-means clusters on 440 million examples by two orders of mangnitude and the time complexity for computing clusters becomes $O(i.m.k^2)$.

After computing cluster centroids on part of the data we assign all examples to their clutsers in $O(n.k)$ time. The overall clustering time complexity is $O(n.k)$ when $i.m.k < n$ and $O(i.m.k^2)$ otherwise. This complexity is lower than $O(n^2)$ as long as $k < \frac{n}{\sqrt{i.m}}$.

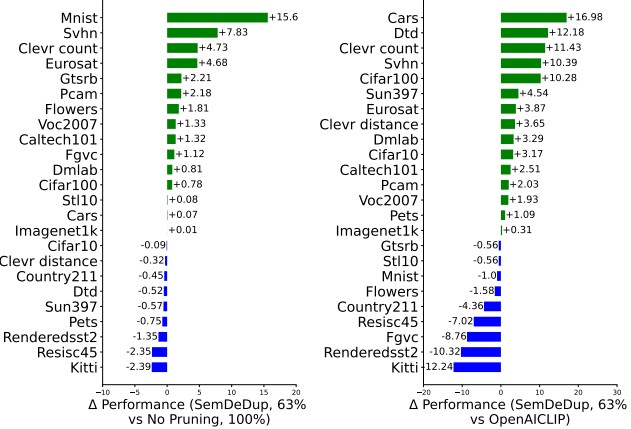

Figure A1: **SemDeDup improves zeroshot and OOD performance in many tasks (the green bars) with less pre-training.** A comparison of zeroshot evaluation performance between our CLIP model trained on 63% of LAION-440M after de-duplication to a baseline CLIP model trained on 100% of the data (**left**), and an OpenAI CLIP (Radford et al., 2021a) (**right**) on 24 tasks.

## A.2    Number of k-means Clusters for SemDeDup

To further assess the impact of changing the value of $k$ we measure the intersection between datasets deduplicated by SemDeDup using different values for $k$. Let $D_A = \{a_1, a_2, ..., a_N\}$ and $D_B = \{b_1, b_2, ..., b_N\}$ be two datasets of the same size $N$. We define the percentage of intersection $I$ between $D_A$ and $D_B$ in equation 1 as the percentage of data points that appear in both datasets relative to the dataset size $N$. Note that $I(D_A, D_A) = 100\%$.

We find that deduplicating LAION-440M dataset to 72% of its size using any value of $k$ values (10000, 25000, 50000, 70000) results in almost the same dataset with only 3% of the examples replaced when changing $k$. This is induced by the 97% percentage of intersection $I$ value between any pair of datasets deduplicated

using two different values for $k$. We show in Fig. A2 the percentage of intersection ratio between different datasets when changing the number of clusters $k$ at different deduplication thresholds $\epsilon$.

We also show in figure A3 that by using the same deduplication threshold value $\epsilon$ we get almost the same deduplicated dataset size for different values for $k$.

$$I(D_A, D_B) = 100 * \frac{|D_A \cap D_B|}{N} \tag{1}$$

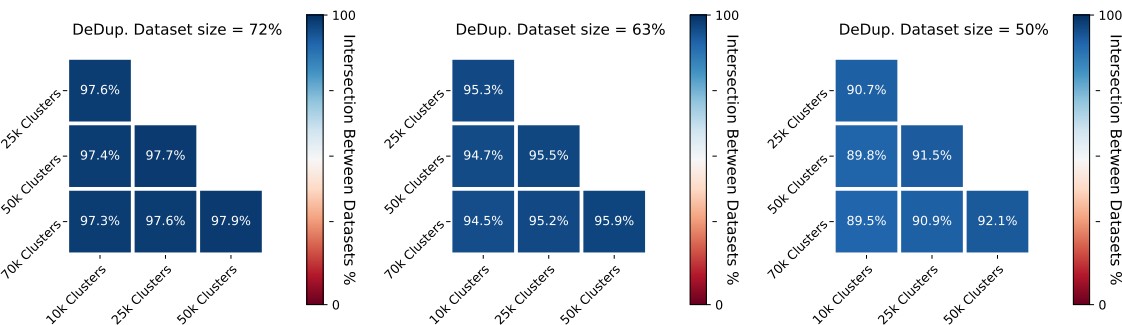

Figure A2: Intersection between different deduplicated LAION datasets using different values for the number of k-means clusters $k$. Each cell corresponds to the percentage of intersection between two datasets deduplicated using different $k$ values. At the 72% dataset size, more than 97% of data examples are shared between all the datasets regardless of the value of $k$. This shows the robustness of SemDeDup to the number of clusters parameter $k$.

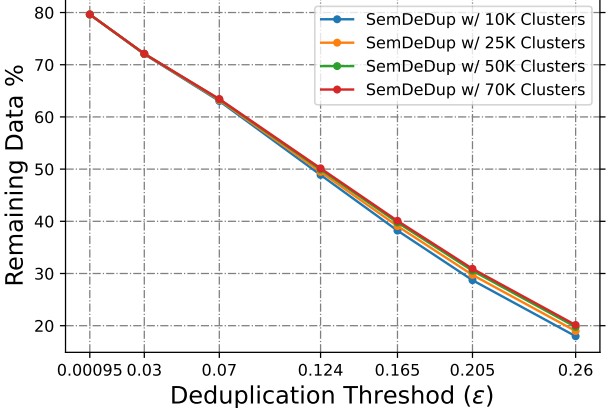

Figure A3: Deduplicated dataset size as a function of the deduplication threshold for different values of k-means clusters $k$. Note that the range in the resulting deduplicated dataset size is 0.003% when $\epsilon$ is 0.00095 and 2% when $\epsilon$ is 0.26.

## A.3 Estimating The Fraction of Duplicates Detected By SemDeDup

SemDeDup searches for duplicates within clusters. This results in reducing the floating point operations (FLOPs) required for deduplication by 5 order of magnitude for LAION-440M dataset as described in Section

Table A1: By training on only 50% of LAION440M, deduplicated using SemDeDup, we perform better than training on whole LAION440M (Baseline100) with 62.5% or 75% of the number of training iterations used for training the baseline model. The table shows zeroshot Top1 accuracy.

| Model / Metric | IN Acc | Avg. Acc (24 datasets) | Avg. OOD (6 datasets) |
|---|---|---|---|
| 100% data, 100% iters (Baseline100) | 68.65 ±0.10 | 54.17 ±0.65 | 55.84 ±0.12 |
| 50% data, 50% iters | 68.27 | 52.44 | **55.87** |
| 50% data, 62.5% iters | 68.33 | **55.07** | **56.38** |
| 50% data, 75% iters | **69.21** | **55.07** | **56.36** |

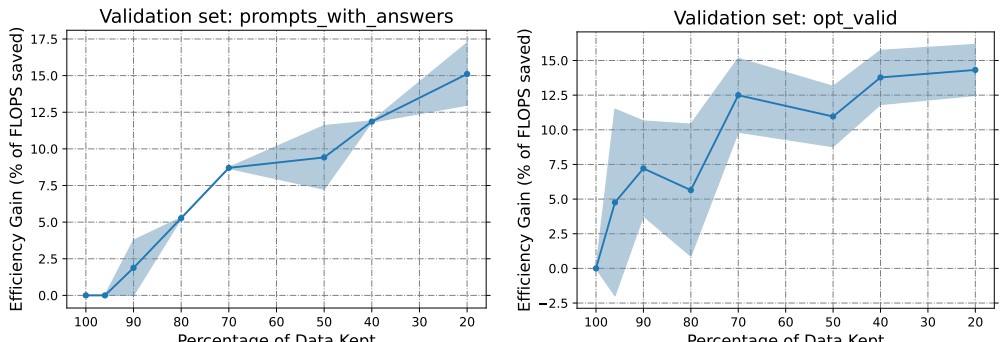

Figure A4: **SemDeDup allows compute efficiency gains by training on much smaller datasets for slightly longer.** We prune datasets via SemDeDup and continue training past one epoch until we reach baseline model perplexity. The x-axis is the percentage of data kept, and the y-axis is the percentage of FLOPs saved. For example, training on the 80% pruned dataset reaches baseline model perplexity on prompts_with_answer in 95.0% of the baseline training, saving 5.0% compute. Mean and standard deviation provided across 3 random training seeds.

3. Indeed, by searching for duplicates within clusters, we ignore duplicates across different clusters if they exist. Here we try to estimate the efficiency of SemDeDup in detecting all the duplicates in the dataset.

Let $D_\epsilon$ represent the total number of duplicates in the dataset at a specific value of deduplication threshold $\epsilon$, and $D_\epsilon^s$ represent the total number of duplicates detected by SemDeDup. We define the deduplication efficiency $\eta_\epsilon$ (eq. 2) as the fraction of duplicates detected by SemDeDup from the total number of duplicates in the datasets at a specific value of $\epsilon$. For example, a deduplication efficiency of 100% corresponds to detecting all the duplicates in a dataset. As computing the exact value of $D_\epsilon$ is computationally expensive, we approximate its value by the number of duplicates between the cluster items and its 20 nearest neighbor clusters and donate this approximated value by $D_\epsilon^{'}$. We sampled part (2000 clusters) of the LAION-440M dataset randomly and compute the value of the deduplication efficiency $\eta$ in eq. 2 for different values of $\epsilon$ and k-means clusters $k$. As we see in Table A2, for $k$=50,000, SemDeDup can effectively detect more than 94% of the duplicates when keeping 63% of LAION-440M dataset and 89% of the duplicates when keeping 40%.

$$\eta = 100 * \frac{D_\epsilon^s}{D_\epsilon^{'}} \tag{2}$$

## B   Using image hashing algorithms for measuring semantic similarity:

We studied the relationship between other baselines like hashing methods and SemDeDup. Our evaluation, using Difference Hash (dhash), (Chamoso et al., 2017), as an example, indicates that these methods are

Table A2: Percentage of duplicates detected ($\eta$) by SemDeDup at different deduplication thresholds ($\epsilon$). We notice that $\eta$ increases as we reduce the number of clusters $k$ in the clustering step of SemDeDup.

| Percentage of Data Kept | 63% | | | 50% | | | 40% | | |
|---|---|---|---|---|---|---|---|---|---|
| Num. of Clusters | 70K | 50K | 10K | 70K | 50K | 10K | 70K | 50K | 10K |
| $\eta$ | 94.4 | 94.6 | 95.3 | 90.1 | 90.6 | 91.3 | 88.3 | 89.0 | 90.8 |

Table A3: The effectivness of using image similarity comparisons in raw pixel space.

| Method / Metric | SemDeDup | Clustering+Dhash | Random Pruning |
|---|---|---|---|
| Top1 zeroshot IN Acc. (Dataset Kept %) | **66.90%** (40%) | 64.10% (42%) | 64.80% (40%) |
| Top1 zeroshot IN Acc. (Dataset Kept %) | **68.62%** (63%) | 67.24% (63%) | — |

unlikely to perform as effectively as SemDeDup in detecting semantic duplicates.

In order to assess the effectiveness of using image hashing algorithms (dhash), we replace the within-cluster cosine similarity search in SemDeDup with a within-cluster Hamming distance search conducted on dhash codes derived from LAION-440M image pixels (we hash the images in the LAION-440M dataset using the *imagehash* library into binary hash codes of a fixed length of 256). We train two models using 42% and 63% of LAION-400M, respectively. We find that SemDeDup with cosine similarity outperforms both models, when using 40% and 63% of LAION-440M, respectively. We show the result table A3 which demonstrates the challenges of performing image similarity comparisons in raw pixel space, rather than using pre-trained embeddings.

**Why dhash is not a good metric for semantic image similarity in contrast to cosine similarity?** We find that the intersection between examples removed by dhash and cos similarity is very small on the LAION-440M dataset as shown if Fig. A5. To measure the correlation between using cosine similarity between embeddings and Hamming distance between dhash image code computed from image pixels as a metric for similarity we sampled a small subset from the LAION-440M dataset and computed 2 million similarity values between them. We found that the Pearson's correlation between both metrics is 0.051.

## C  Choosing the deduplication threshold $\epsilon$

We tune the deduplication threshold $\epsilon$ for each dataset manually to get the desired deduplicated dataset size. To do that, we first run the clustering step of SemDeDup. Then we sample 10% of the clusters and tune $\epsilon$ on them. We found that using only 10% of clusters gives a good approximation of the final dataset size. We notice that the relationship between $\epsilon$ and the deduplicated dataset size is semi-linear for both LAION and C4 datasets (see Fig. 3, A3, and A21). When tuning $\epsilon$, we start with two values and run SemDeDup on 10% of the clusters (the time needed for this step is a few minutes. See the DeDup. Time column in Table A5 ). Then we linearly interpolate the two values of $\epsilon$ knowing their correspondence deduplicated dataset sizes and the target dataset size to get a better value for $\epsilon$. In Fig. A3 we plot the duplicated dataset size as a function of $\epsilon$ for different values of the number of clusters $k$ used. We show that $k$ has a small impact on the value of $\epsilon$ only when the duplicated dataset size is less than 50%.

## D  Compute cost of running SemDeDup

We report in Table A4 the cost of running SemDeup on LAION440M in GPU hours. We see in the table that the overhead of deduplicating LAION440M doesn't exceed 1% of the training cost in GPU hours. This results in substantial savings in the overall cost after deduplication. For example, training on 50% of the data saves 50% of the training cost while requiring only 1% of the training cost for deduplication. We also show in

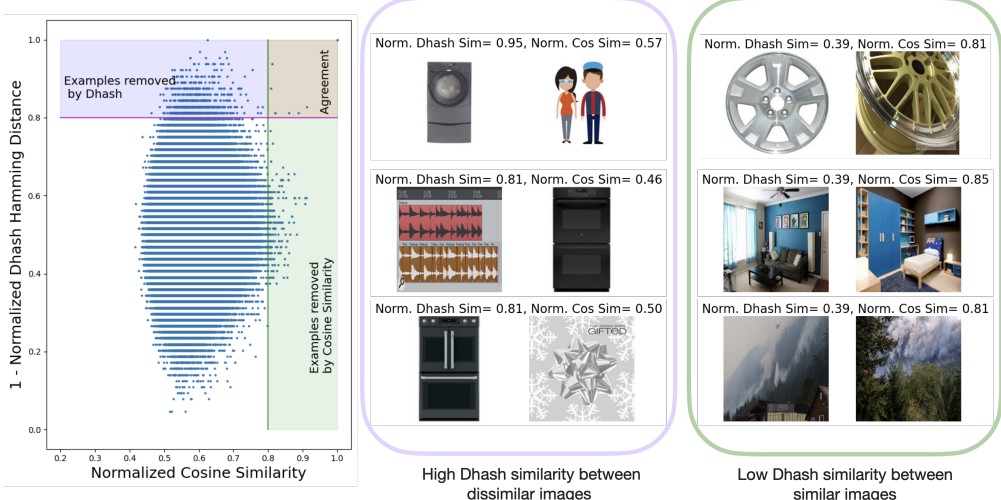

Figure A5: The plot compares the similarity measures between image smaples from LAION-440M dataset using 1) cosine similarity in the embeddings space and 2) Hamming distance between dhash codes computed from image pixels. Hamming distance is normalized from [0-64] to [0-1] (a lower value indicates high similarity). Cosine similarity is normalized to be between [0-1] as well. The plot shows (1 - Hamming distance) to maintain consistency with cosine similarity. The plot reveals a very weak correlation between cosine similarity and Dhash Hamming distance with a Pearson's correlation of 0.051.

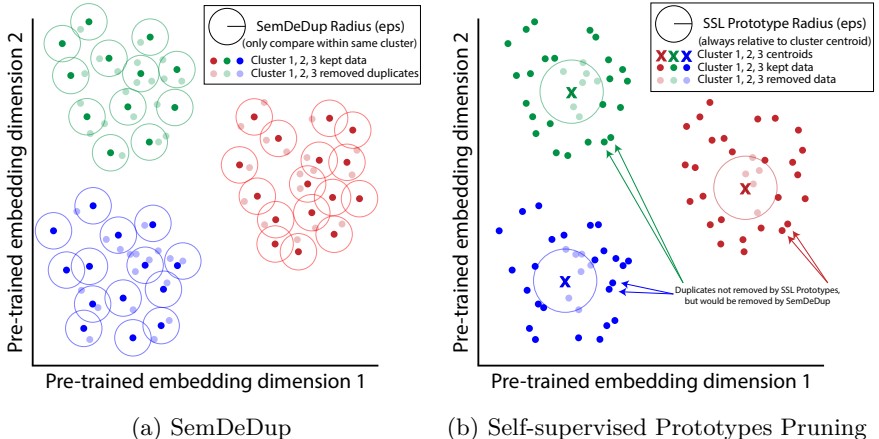

        (a) SemDeDup         (b) Self-supervised Prototypes Pruning

Figure A6: We compare between SemDeDup and the Self-Supervised Prototypes (SSP) Pruning method of (Sorscher et al., 2022). While both methods cluster the data in the embedding space, they differ in the way remove examples. SemDeDup looks at pairwise similarity between cluster items. SSP ranks examples by their distance to to cluster centroids.

Table A5 the time needed for deduplicating LAION440M dataset using SemDeDup. Our implementation for SemDeDup parallelizes the operations across devices to speed up the deduplication. Table A5 also shows how the deduplication time changes as we change the number of clusters.

However, we should note that the computational cost of SemDeDup can be amortized across the efficiency gains it can generate in training many downstream models by many other groups. For example, its typical use case would be to take a large web-scaled dataset, and semantically deduplicate it *once*, resulting in a much smaller *foundation dataset* (Sorscher et al., 2022) that can be widely disseminated to the community. Then many different groups can train many different foundation models on this deduplicated foundation

Table A4: SemDeDup requires much fewer GPU hours than training a CLIP ViT-B-16 model on LAION440M for one epoch. When using 50K clusters, it requires only 0.29 of the GPU hours needed for one epoch of training on 100% of the data. This is equivalent to 0.0091 of the complete training cost in GPU hours.

| Num. SemDeDup Clusters / Cost | GPU Hours For Training CLIP on 100% of LAION440M for 32 Epochs | GPU Hours For Training on 50% of DeDup. Data | SemDeDup Overhead in GPU Hours | SemDeDup Overhead / 1 Epoch Training GPU Hours | SemDeDup Overhead / 32 Epochs Training GPU Hours |
|---|---|---|---|---|---|
| 10K Clusters | 11541 | 5770.5 | 163.5 | 0.43 | 0.0132 |
| 25K Clusters | 11541 | 5770.5 | 101.2 | 0.26 | 0.0082 |
| 50K Clusters | 11541 | 5770.5 | 110.3 | 0.29 | 0.0091 |
| 70K Clusters | 11541 | 5770.5 | 103.6 | 0.27 | 0.0084 |

Table A5: Time for running SemDeDup on LAION440M. Note that we report the total time for deduplication to different dataset size ratios.

| Operation / Time | Clustering Time | DeDup. Time | Total Time |
|---|---|---|---|
| SemDeDup w/10K Clusters | 2h:36 @8 GPUs | 2h:20 @64 GPUs | 4h:56 |
| SemDeDup w/25K Clusters | 3h:52 @8 GPUs | 1h:19 @64 GPUs | 5h:11 |
| SemDeDup w/50K Clusters | 5h:59 @8 GPUs | 1h:22 @64 GPUs | 7h:21 |
| SemDeDup w/70K Clusters | 9h:02 @8 GPUs | 1h:10 @64 GPUs | 10h:12 |
| Training CLIP on 100% of LAION440M for 32 Epochs | — | — | 69h:52 @176 GPUs |

dataset, and all these groups will reap the training efficiency gains conferred by a less redundant smaller dataset. Thus the computational cost of finding the dataset can be amortized across the efficiency gains achieved on many downstream training runs, in direct analogy to how the computational cost of training a foundation model can be amortized across the computational efficiency gains with which it achieves high zero-shot or fine-tuning performance on many downstream applications.

# E  CLIP Zeroshot Evaluation

In this Section, we show the result of zeroshot evaluation for CLIP. We note that the models trained on dataset deduplicated using SemDeDup outperform the baseline model in many tasks. In Table A7 we list the top1 zeroshot accuracy on 24 tasks and in Table A8 we show the top1 zeroshot accuracy on 6 datasets for out-of-distribution robustness evaluation. Our complete evaluation set has 30 different datasets in total. When using only 63% of LAION-440M, SemDeDup outperforms the baseline model in 19 out of the 30 tasks. Fig. (A8) and Fig. (A9) show the performance of different models as a function of training dataset size.

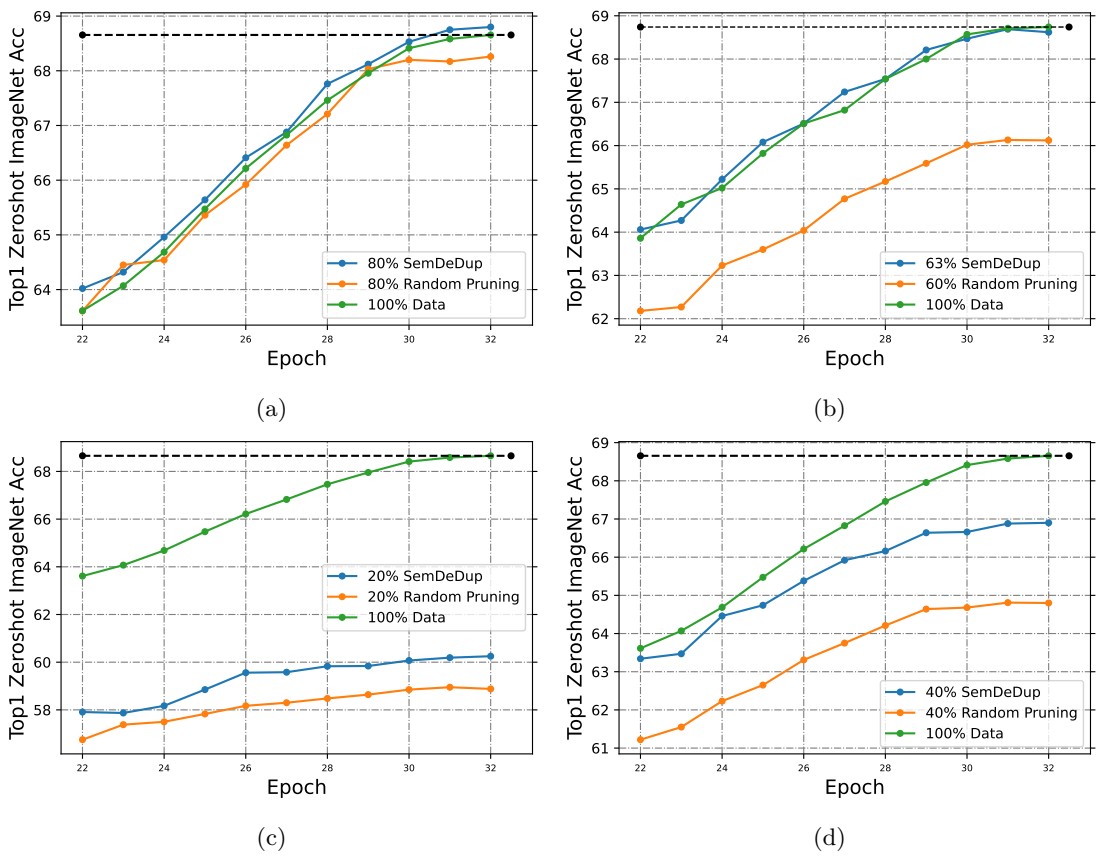

Figure A7: SemDeDup is always better than training on random subset from LAION-440M. The plots show zeroshot top1 accuracy on ImageNet for CLIP models trained on different fractions of data.

Table A6: Training parameters for CLIP

| Parameter | Value |
|---|---|
| Model | CLIP ViT-B-16 |
| Warmup | 2000 training steps |
| Epochs | 32 |
| Batch size | 33,792 |
| Learning rate | 5.0e-4, cosine scheduler |
| Optimizer | AdamW, wd=0.2, betas=(0.9, 0.98), eps=1.0e-6 |

# F   LAION-233M De-duplication

To support our results on LAION-440M, we also de-duplicate a much smaller dataset of 233 million images. We call this dataset LAION-233M. Usually, CLIP needs to be trained on more than 400 million image-caption pairs as introduced in (Radford et al., 2021a), so de-duplicating LAION-233M is more challenging in this respect. We train a baseline model on the 233 million image-caption pairs and two models on 55% of the data, one on a random subset and the other on deduplicated subset using SemDeDup. We trained all the models using the same hyperparameters we used for training on LAION-440M. We show ImageNet top1

Table A7: **Zeroshot evaluation** top1 accuracy on different datasets. Training CLIP on 72% of the data after deduplication gives a higher performance than training on 100% of the data in 19/24 datasets. In the first row, model names are represented by the pruning method (Dedup, Baseline, and Rand for SemDeDup, no pruning, and random pruning respectively), and the fraction of data used for training. For the Baseline100 model we report the average performance across 4 training seeds.

| Data / Model | Dedup20 | Dedup40 | Dedup50 | Dedup63 | Dedup72 | Dedup80 | Baseline100 | Rand80 | Rand60 | Rand40 | Rand20 |
|---|---|---|---|---|---|---|---|---|---|---|---|
| Cars | 63.33 | 78.14 | 80.26 | 81.43 | 82.05 | **82.61** | 81.37 | 80.96 | 79.26 | 77.74 | 71.57 |
| Country211 | 14.16 | 17.74 | 18.26 | 18.44 | 18.70 | 18.20 | **18.88** | 18.39 | 16.75 | 15.97 | 12.88 |
| Fgvc Aircraft | 4.44 | 11.49 | 12.42 | **15.42** | 15.27 | 15.09 | 14.30 | 14.31 | 13.11 | 9.27 | 8.85 |
| GTSRB | 38.22 | 37.20 | 36.22 | 43.06 | 41.00 | 35.74 | 40.85 | **43.33** | 41.88 | 25.72 | 32.28 |
| Imagenet1k | 60.24 | 66.90 | 68.27 | 68.66 | **68.93** | 68.80 | 68.65 | 68.29 | 66.12 | 64.82 | 58.86 |
| MNIST | 44.29 | 31.87 | 22.93 | 48.55 | 42.75 | **48.86** | 32.95 | 43.82 | 35.73 | 36.32 | 19.22 |
| Renderedsst2 | 51.46 | 53.65 | 52.72 | 50.80 | 52.99 | **57.17** | 52.14 | 52.72 | 51.29 | 52.22 | 45.47 |
| STL10 | 96.06 | 96.85 | 97.50 | **97.71** | 97.69 | 97.21 | 97.63 | 97.49 | 97.38 | 97.08 | 94.31 |
| SUN397 | 64.81 | 67.98 | 68.26 | 68.89 | 69.25 | **69.76** | 69.46 | 69.08 | 67.96 | 65.51 | 60.76 |
| VOC2007 | 77.94 | 79.51 | 79.74 | **80.37** | 79.75 | 78.61 | 79.04 | 77.97 | 79.43 | 77.96 | 74.42 |
| Caltech101 | 83.05 | 84.40 | 84.98 | **85.06** | 84.35 | 84.75 | 83.74 | 83.69 | 83.93 | 83.38 | 80.62 |
| CIFAR100 | 72.09 | 75.71 | 75.17 | **77.19** | 77.16 | 77.08 | 76.41 | 76.02 | 74.08 | 72.37 | 67.79 |
| CIFAR10 | 92.80 | 93.78 | 94.01 | 94.00 | **94.49** | 94.13 | 94.09 | 94.25 | 93.95 | 92.68 | 89.14 |
| Clevr Dist | 15.75 | **23.05** | 15.75 | 19.48 | 21.95 | 21.82 | 19.80 | 18.45 | 15.59 | 18.60 | 16.21 |
| Clevr Count | 25.37 | 26.36 | 30.85 | 31.87 | **34.73** | 20.31 | 27.14 | 15.37 | 26.43 | 14.85 | 21.67 |
| DMLAB | 13.16 | 17.62 | 17.99 | 19.20 | 18.52 | 20.23 | 18.39 | 18.50 | **21.05** | 17.12 | 19.36 |
| DTD | 49.73 | 53.51 | 54.31 | 56.76 | **58.94** | 57.66 | 57.27 | 57.02 | 53.35 | 50.96 | 41.76 |
| Eurosat | 44.07 | 51.28 | 51.70 | 59.46 | 57.02 | 59.72 | 54.78 | **59.81** | 48.63 | 51.26 | 50.00 |
| Flowers | 45.21 | 62.21 | 67.67 | 69.78 | **70.48** | 66.29 | 67.97 | 68.39 | 65.43 | 62.42 | 58.16 |
| Kitti Dist | 20.39 | 13.36 | 14.35 | 14.77 | 19.97 | **26.72** | 17.16 | 11.11 | 20.68 | 17.02 | 11.25 |
| PCAM | 49.69 | 48.83 | 47.62 | **52.66** | 50.09 | 52.14 | 50.48 | 50.11 | 41.28 | 55.02 | 56.59 |
| Pets | 77.87 | 87.30 | 89.72 | 90.02 | 90.16 | 90.57 | **90.77** | 90.49 | 89.86 | 88.72 | 82.50 |
| Resisc45 | 46.76 | 57.56 | 51.69 | 51.49 | 50.14 | 53.57 | 53.84 | **54.06** | 51.65 | 49.29 | 46.72 |
| SVHN | 34.80 | 33.64 | 26.24 | **40.87** | 33.96 | 32.68 | 33.04 | 26.77 | 26.67 | 32.70 | 25.26 |
| Average | 49.4 | 52.91 | 52.44 | **55.66** | 55.43 | 55.41 | 54.17 ±0.65 | 53.77 | 52.56 | 51.21 | 47.73 |

Table A8: **Out-of-distribution Robustness** for CLIP models we trained on a different number of examples. The two models trained on 63% and 72% of LAION440M with our de-duplication method have higher average accuracy over 6 datasets. In the first column, model names are represented by the pruning method (Dedup, Baseline, and Rand for SemDeDup, no pruning, and random pruning respectively), and the fraction of data used for training. For the Baseline100 model we report the average performance across 4 training seeds.

| Model/-Dataset | ImageNet-A | ImageNet-O | ImageNet-R | ImageNet-Sketch | ImageNet-V2 | Ob-ject-Net | Average |
|---|---|---|---|---|---|---|---|
| Dedup20 | 31.35 | 52.25 | 72.69 | 46.98 | 52.71 | 51.0 | 51.16 |
| Dedup40 | 38.73 | 49.3 | 77.08 | 51.93 | 59.21 | 54.98 | 55.21 |
| Dedup50 | 39.68 | 48.55 | 77.74 | 53.54 | 60.37 | 55.36 | **55.87** |
| Dedup63 | 39.07 | 48.45 | 78.24 | 53.86 | 60.56 | 56.33 | **56.08** |
| Dedup72 | 39.53 | 47.6 | 78.61 | 53.7 | 61.23 | 56.28 | **56.16** |
| Dedup80 | 39.12 | 47.95 | 78.53 | 53.82 | 60.59 | 54.72 | 55.79 |
| Base-line100 | 38.56 | 47.59 | 78.65 | 53.81 | 60.96 | 55.44 | 55.84 ±0.12 |
| Rand80 | 37.87 | 47.7 | 78.04 | 52.81 | 60.02 | 54.3 | 55.12 |
| Rand60 | 34.6 | 47.5 | 75.61 | 51.18 | 57.97 | 53.22 | 53.35 |
| Rand40 | 31.88 | 49.1 | 73.65 | 49.02 | 56.83 | 49.57 | 51.67 |
| Rand20 | 23.43 | 49.4 | 66.74 | 43.76 | 50.67 | 43.57 | 46.26 |

Table A9: Performance after training on deduplicated data for the same number of examples seen as training on 100% of the data.

| Metric / Model | Dedup40 | Dedup50 | Dedup60 | Dedup70 | Baseline (100%) |
|---|---|---|---|---|---|
| Top1 IN Zeroshot Acc. | 68.35 | **68.92** | **69.04** | **69.14** | 68.65 ±0.10 |
| Top5 IN Zeroshot Acc. | **91.64** | **91.82** | **91.86** | **91.73** | 91.46 ±0.04 |

zeroshot accuracy for these models in Fig. A10. The baseline model achieved 64.62% accuracy, while the SemDeDup model achieved 63.61% outperforming the model trained on the random subset (61.3% accuracy).

# G  Visualizing Examples Before and After De-duplication

To visually show which images are removed by SemDeDup from LAION-440M dataset, we visualize some images from a random cluster before and after deduplication. To do that, we choose a cluster randomly and sort its examples by the cosine similarity to the centroid. By doing that, we can show similar images next to each other in a sequence. Then we visualize a sequence of images before de-duplication. After that, we run SemDeDup, remove duplicates, and sort the remaining examples again. Finally, we visualize the sequence of images from the same indices we visualize before de-duplication. We visulaize some examples of clusters before and after deduplication in Fig. (A14 and A15). The figures show that after applying SemDeDup with different values for the de-duplication threshold $\epsilon$, we keep the unique images.

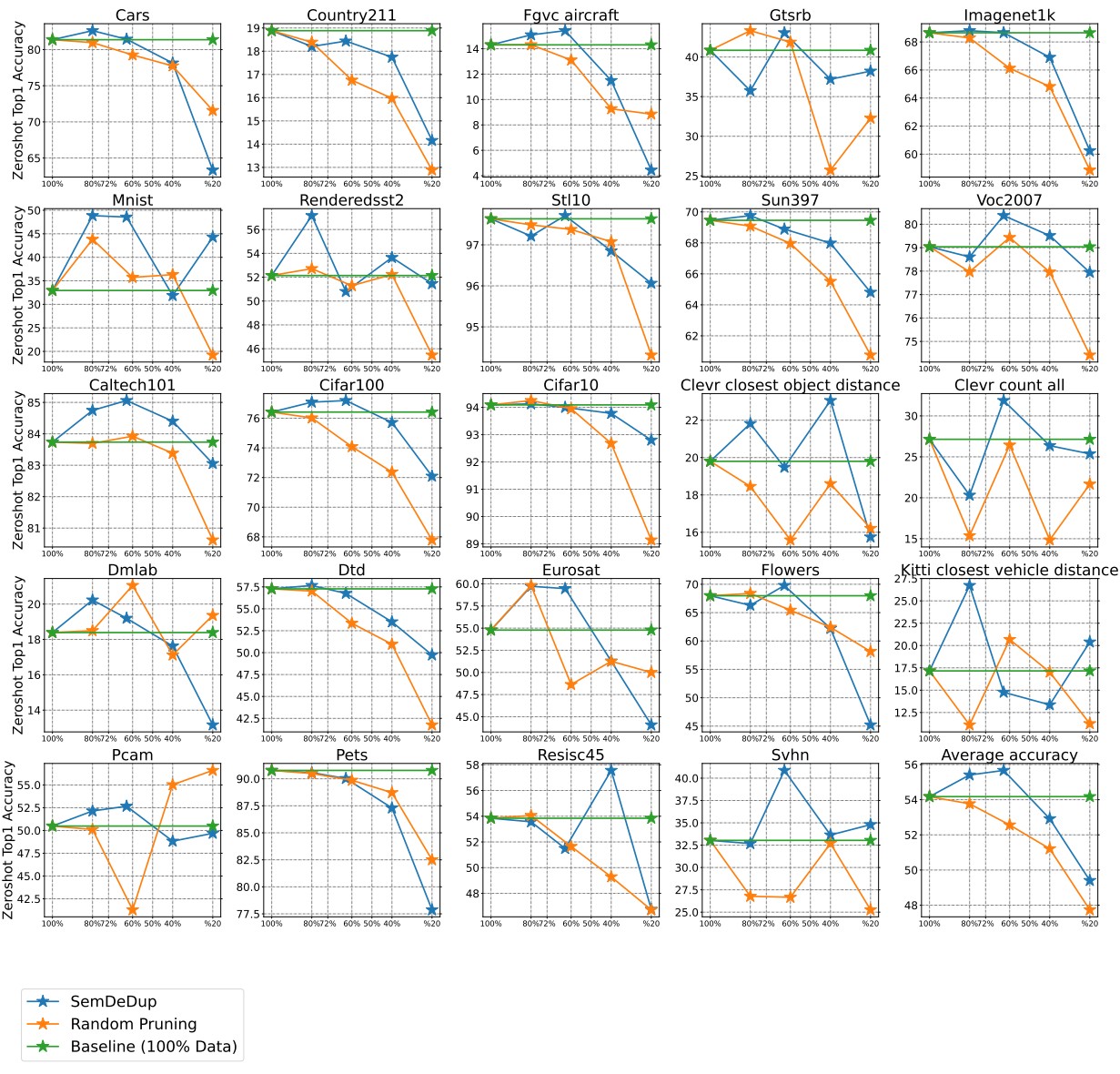

Zeroshot Top1 Accuracy

Figure A8: Zeroshot performance of CLIP on 24 datasets. The last plot shows the average performance over all datasets.

## H   Perplexity Values for SemDeDup on Language Modeling

## I   What is being pruned in language data?

In Fig. A26 and Fig. A27 we choose specific clusters and show a random sample of documents retained in the cluster after performing SemDeDup for different values of $\epsilon$. In Fig. A26, we observe that at low values of $\epsilon$, we find semantic duplicates in the form of templated text, where typically few words (e.g. a geographic location or a name) is changed. This successfully evades exact-string deduplication methods but contains highly redundant information as seen in Fig. A26. In Fig. A27, we show an example of a cluster

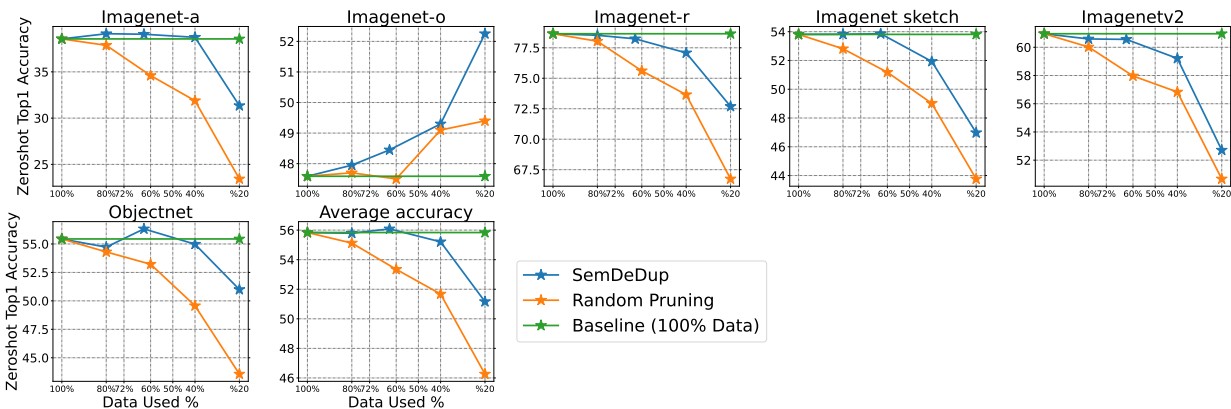

Figure A9: Out-of-distribution zeroshot performance on 6 datasets. SemDeDup outperforms random pruning on all datasets for all fractions of dataset kept. The last plot shows the average performance over all datasets.

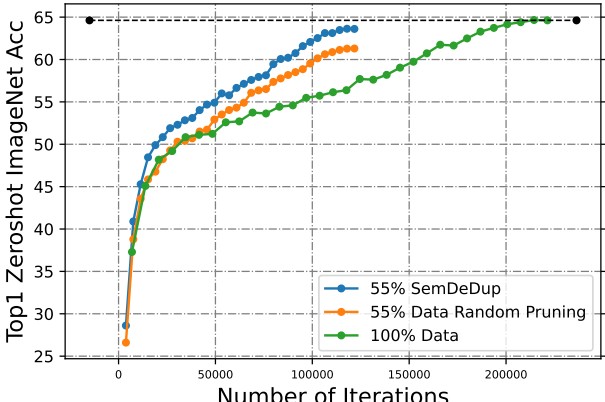

Figure A10: Performance when deduplicating 233 million images from LAION-2B. We deduplicte LAION233M to 55% of its size and train CLIP model on it. SemDeDup performs better than random pruning (63.61% vs 61.3%). Training on the whole 233 million examples gives 64.62%. Note that the deduplicated dataset size here is 128 million only.

with semantically redundant duplicates — most examples in this cluster are advertisements about Nike shoes. These examples are not necessarily templated text or have exact string matches, but are highly redundant nonetheless. We see in Fig. A27 that at more aggressive pruning (i.e. higher $\epsilon$) these semantically redundant duplicates get pruned. We note that exact string duplicates (i.e.“perceptual duplicates for text") are rare since duplicate occurrences of any three-sentence spans were removed in C4 already.

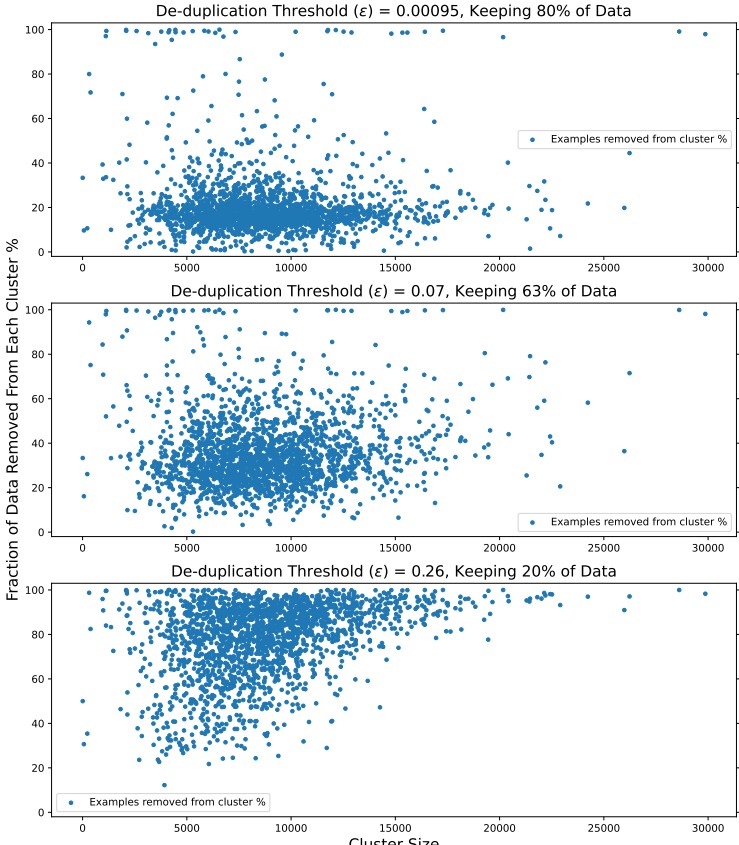

Figure A11: How many images can we remove from each cluster? Moving from top to down we increase $\epsilon$ value. The x-axis corresponds to the cluster size. The y-axis corresponds to the fraction of data removed from each cluster by SemDeDup. As we increase $\epsilon$, more examples are removed from each cluster. We notice that most of the examples from the large clusters (the points to the right) are removed when $\epsilon$ becomes large. The points in this figure are for 2000 clusters sampled randomly from a total of 50,000 clusters.

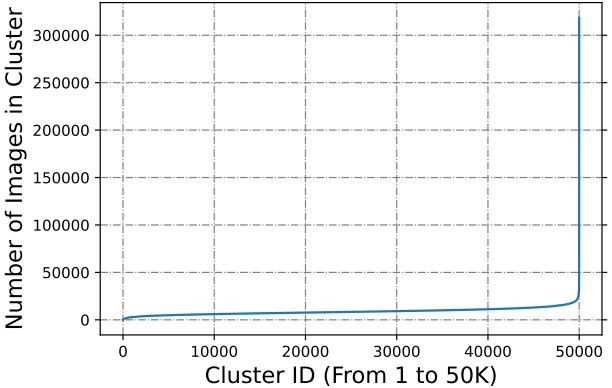

Figure A12: The number of images in each cluster for 50,000 clusters of LAION-440M images after running k-means clustering in the embedding space. The average cluster size is 8748, but we also see a few clusters with more than 300,000 examples.

Table A10

---

PyTorch-style Pseudo Code For SemDeDup

---

```python
#Input: cluster_embeddings, num_clusters, epsilon

for i in range(num_clusters):
    # Load cluster embeddings.
    cluster_i_embeddings = cluster_embeddings[i]

    # Sort the cluster embeddings by the distance to the cluster centroid.
    cluster_i_embeddings = sort_by_distance_to_cluster_centroid(
    cluster_i_embeddings, descending = True)

    # We use descending=True/False for keeping examples with low/high similarity
     to cluster centroids. We  ignore this step for keeping random examples from
     each group of similar examples. See section 6.3 for more details about this
     step.

    # Compute the pairwise cosine similarity between embeddings
    pairwise_sim_matrix = cluster_i_embeddings @ cluster_i_embeddings.T

    triu_sim_matrix = torch.triu(pairwise_sim_matrix, diagonal = 1)

    M = torch.max(triu_sim_matrix, dim=0)[0]

    # Check if the maximum similarity <= the threshold.
    points_to_keep_from_cluster_i = M <= 1-epsilon
```

---

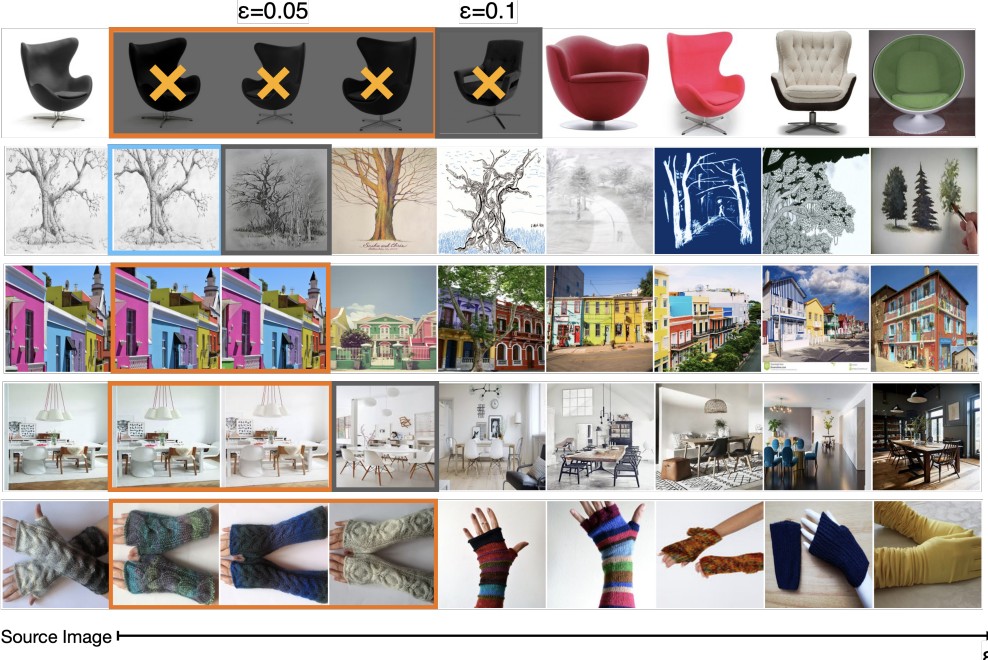

Figure A13: For each of the source images **(left column)**, we can retrieve a set of similar images from LAION-440M. For each of the source images, we show a set of images with the highest cosine similarity to it. Images are sorted from left to right by their cosine similarity (1- $\epsilon$) to the source image. By changing $\epsilon$ value, we can identify perceptual duplicates, semantic duplicates, and semantically redundant examples for the source images. As we see in the **first row**, by increasing $\epsilon$ we can remove many examples that are semantically similar to the source image.

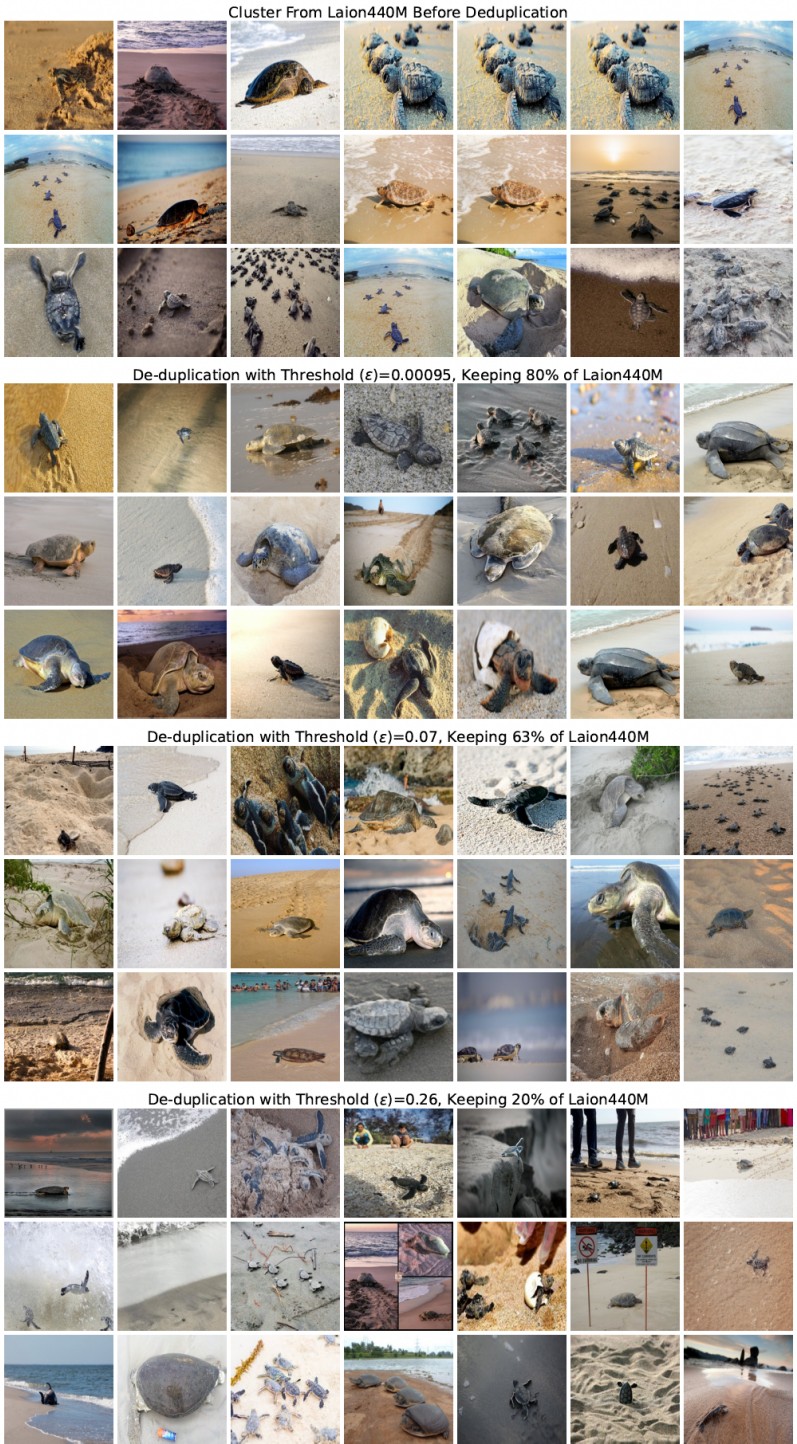

Figure A14: Examples from the same cluster from LAION-440M dataset before and after de-duplication. Images are sorted by cosine similarity to the cluster centroid. Exact duplicates and Near-duplicates are removed at small

$\epsilon$

value. Semantic duplicates are removed as we increase $\epsilon$.

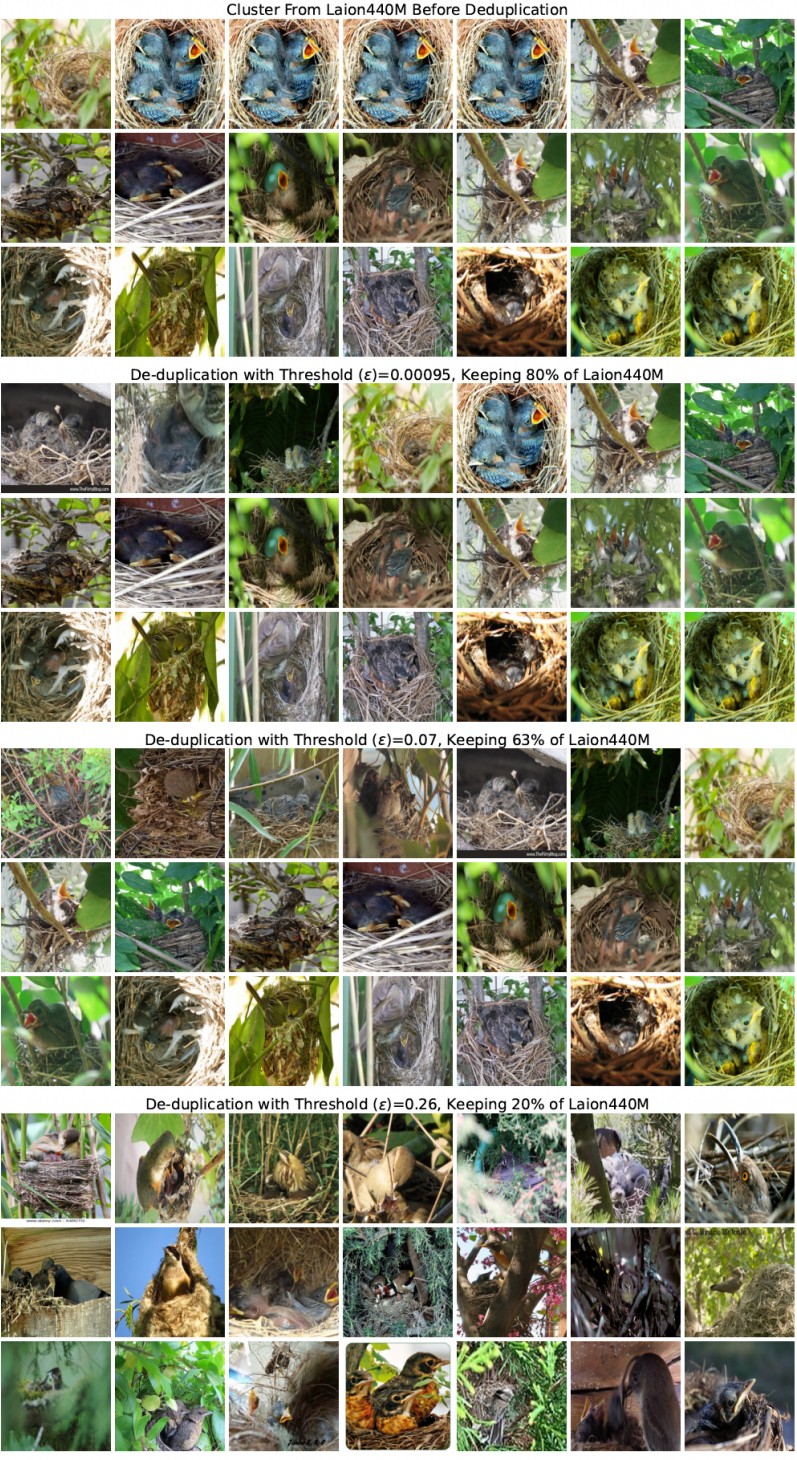

Figure A15: Examples from the same cluster from LAION-440M dataset before and after de-duplication. Images are sorted by cosine similarity to the cluster centroid. Exact duplicates and Near-duplicates are removed at small $\epsilon$ value. Semantic duplicates are removed as we increase $\epsilon$.

| method | Baseline (no pruning) | NearDup from (Lee et al., 2021a) | Random | SemDedup |
|---|---|---|---|---|
| validation set | | | | |
| C4 | 38.95 +/- 0.07 | 39.46 +/- 0.14 | 39.51 +/- 0.07 | **39.35** +/- 0.16 |
| opt_valid | 47.13 +/- 0.21 | 47.33 +/- 0.20 | 47.75 +/- 0.23 | **47.18** +/- 0.29 |
| prompts_with_answers | 29.60 +/- 0.15 | 29.79 +/- 0.11 | 29.95 +/- 0.11 | **29.69** +/- 0.19 |

Figure A16: Comparison of perplexity values for 125M OPT model after pruning via different methods at 96% pruning. Note that (Lee et al., 2021a) pruned 3.9 % of examples, while above Random and SemDeDup prune 4% of examples. Mean and standard deviation provided across 3 training seeds. Note that the Baseline column does not prune data (which is why the perplexities are lower) and bolded numbers compare between Random, SemDedup, and NearDup.

| method | Baseline (no pruning) | Random | SemDedup |
|---|---|---|---|
| validation set | | | |
| C4 | 38.95 +/- 0.07 | 42.16 +/- 0.03 | **41.98** +/- 0.09 |
| opt_valid | 47.13 +/- 0.21 | 50.66 +/- 0.11 | **49.04** +/- 0.16 |
| prompts_with_answers | 29.60 +/- 0.15 | 31.65 +/- 0.16 | **30.98** +/- 0.13 |

Figure A17: Comparison of perplexity values for 125M OPT model after pruning via different methods at 80% pruning. Mean and standard deviation provided across 3 training seeds. Note that the Baseline column does not prune data (which is why the perplexities are lower) and bolded numbers compare between Random and SemDedup.

| method | Baseline | Random | SemDedup |
|---|---|---|---|
| validation set | | | |
| C4 | 38.95 +/- 0.07 | 87.09 +/- 0.21 | **67.32** +/- 0.16 |
| opt_valid | 47.13 +/- 0.21 | 95.05 +/- 0.31 | **70.17** +/- 0.16 |
| prompts_with_answers | 29.60 +/- 0.15 | 60.63 +/- 1.12 | **43.16** +/- 0.19 |

Figure A18: Comparison of perplexity values for 125M OPT model after pruning via different methods at 20% pruning. Mean and standard deviation provided across 3 training seeds. Note that the Baseline column does not prune data (which is why the perplexities are lower) and bolded numbers compare between Random and SemDedup.

| method | Baseline (no pruning) | NearDup from (Lee et al., 2021a) | Random | SemDedup |
|---|---|---|---|---|
| validation set | | | | |
| C4 | 46.16 +/- 0.00 | 46.85 +/- 0.00 | **46.15** +/- 0.00 | 46.56 +/- 0.00 |
| opt_valid | 55.69 +/- 0.00 | 55.27 +/- 0.00 | 55.20 +/- 0.00 | **54.88** +/- 0.00 |
| prompts_with_answers | 34.04 +/- 0.00 | 33.93 +/- 0.00 | 33.91 +/- 0.00 | **33.83** +/- 0.00 |

Figure A19: Comparison of perplexity values for 1.3b OPT model after pruning via different methods at 96% pruning. Note that (Lee et al., 2021a) pruned 3.9 % of examples, while above Random and SemDeDup prune 4% of examples. Due to compute restrictions we do not provide random seed standard deviations.

| method
validation set | Baseline (no pruning) | Random | SemDedup |
|---|---|---|---|
| C4 | 46.16 +/- 0.00 | 303.71 +/- 0.00 | **108.95** +/- 0.00 |
| opt_valid | 55.69 +/- 0.00 | 347.35 +/- 0.00 | **109.68** +/- 0.00 |
| prompts_with_answers | 34.04 +/- 0.00 | 269.96 +/- 0.00 | **72.64** +/- 0.00 |

Figure A20: Comparison of perplexity values for 1.3b OPT model after pruning via different methods at 20% pruning. Note that the Baseline column does not prune data (which is why the perplexities are lower) and bolded numbers compare between Random and SemDedup. Due to compute restrictions we do not provide random seed standard deviations.

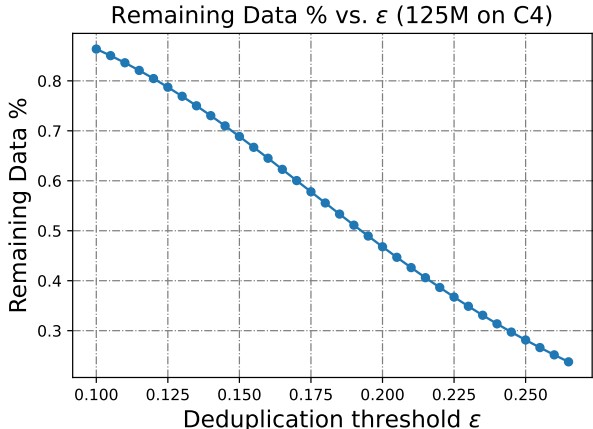

Figure A21: **Percent Data Remaining versus** $\epsilon$ **for C4**. The x-axis corresponds to different values of $\epsilon$ from Section 3, and the y-axis represents the corresponding fraction of data in our subset of C4.

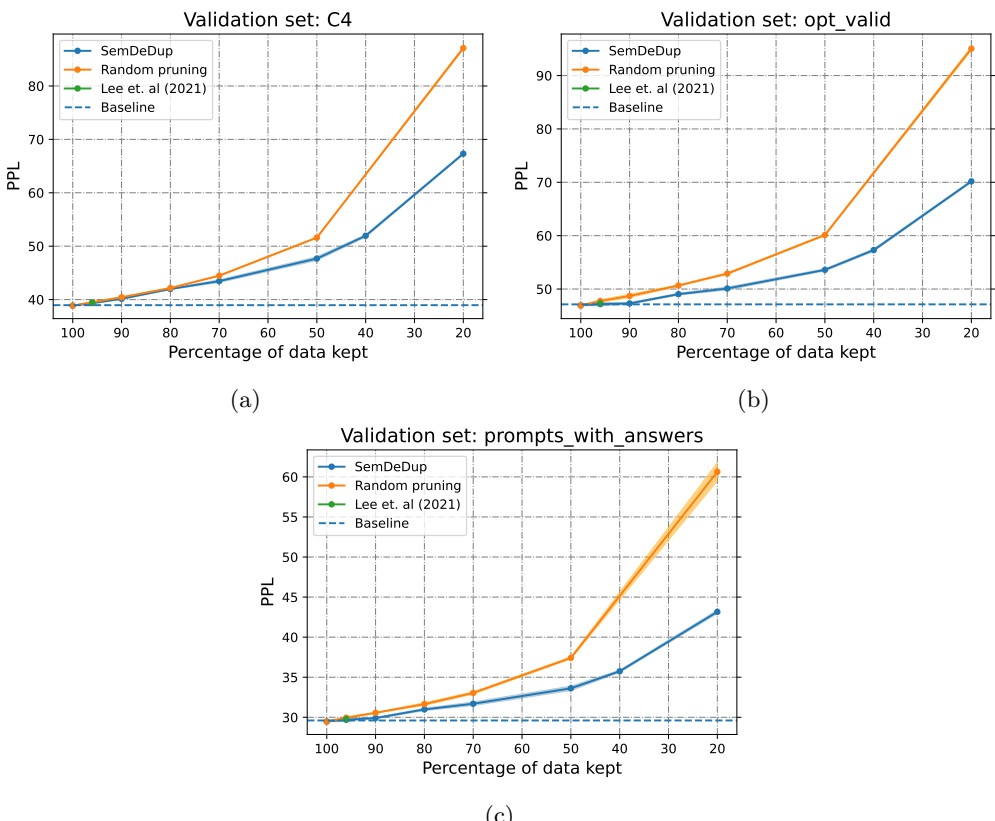

Figure A22: SemDeDup performance at different fractions of data for the 125M OPT model. We show results for the C4 validation set (top left), opt_valid (top right), and prompts_with_answers (bottoms). These are the same graphs as Figure 6, but for a wider range of percentage of data kept. We note that SemDeDup consistently outperforms random pruning at lower percentages of data kept.

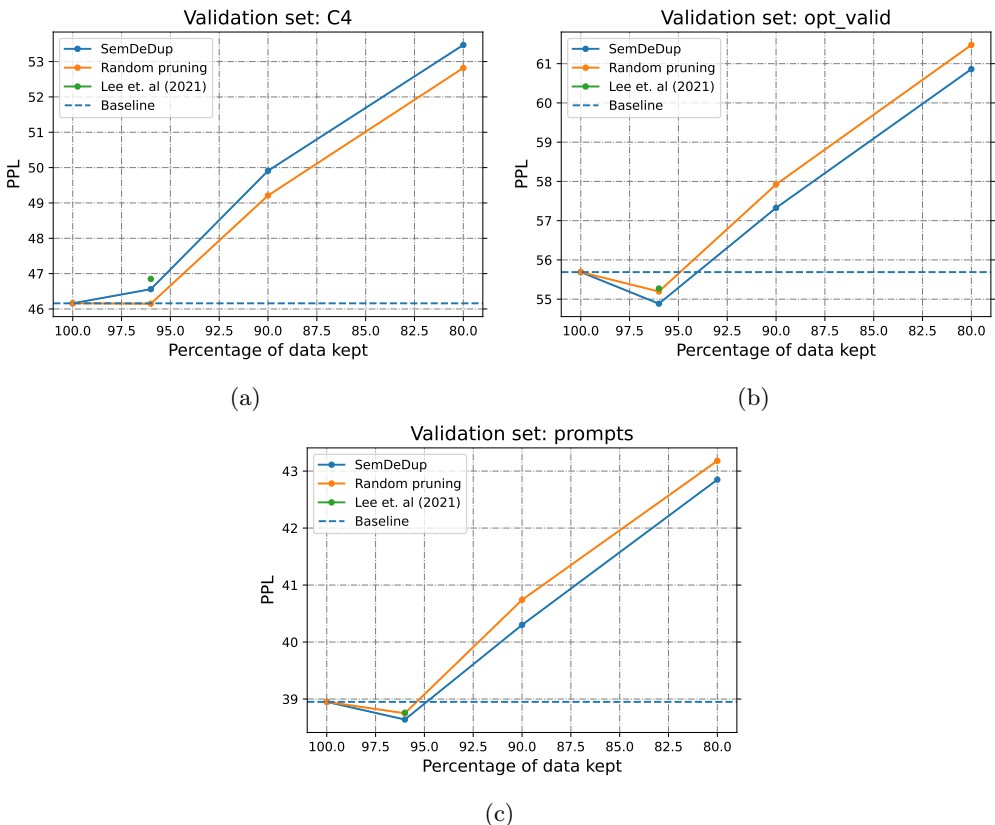

(a)

(b)

(c)

Figure A23: SemDeDup performance at different fractions of data for the 1.3B OPT model. We show results for the C4 validation set (top left), opt_valid (top right), and prompts_with_answers (bottoms). These are similar to tables A19 and A20 but for a range of percentage of data kept (96 %, 90%, 80%). We note that SemDeDup consistently outperforms random pruning at lower percentages of data kept.

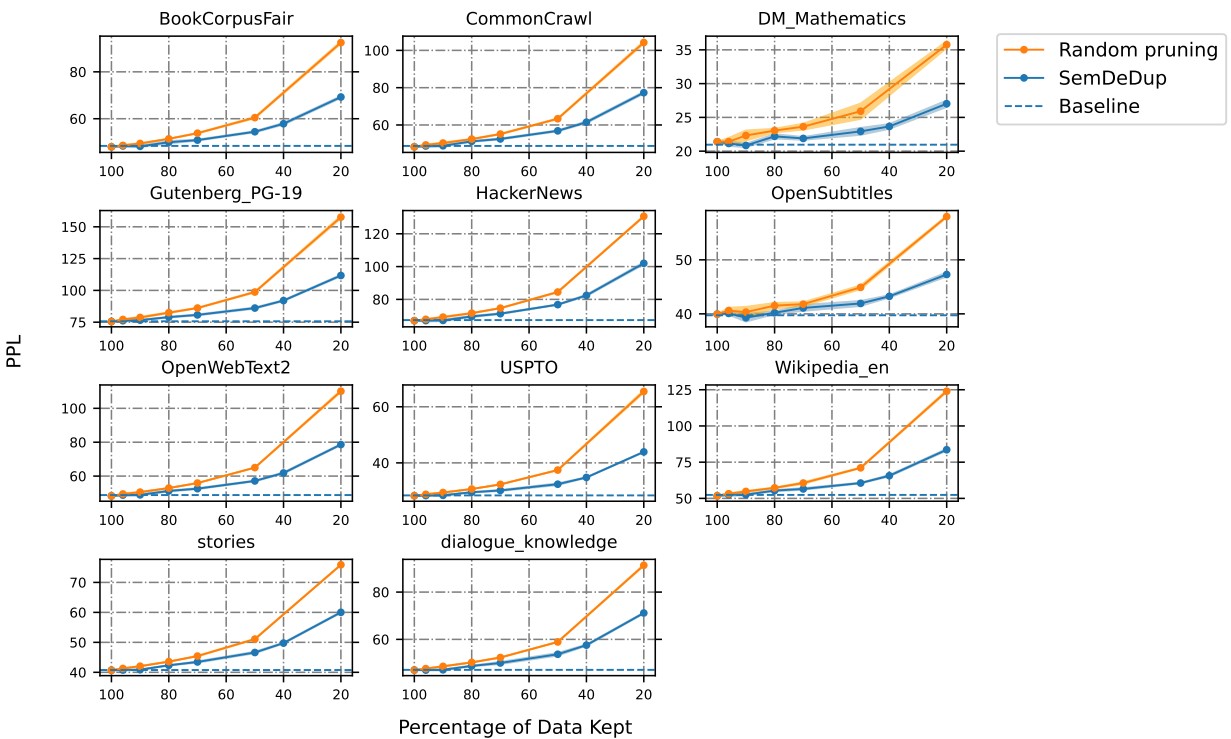

Figure A24: Percentage of Data Kept vs. Perplexity on individual validation sets within opt_valid. Runs are averages across 3 training seeds, and shaded regions represent 1 standard deviation from the mean. The title of each plot represents the name of the individual validation set within opt_valid. Note that on all tasks, SemDedup significantly random pruning, especially at low percentages of data kept.

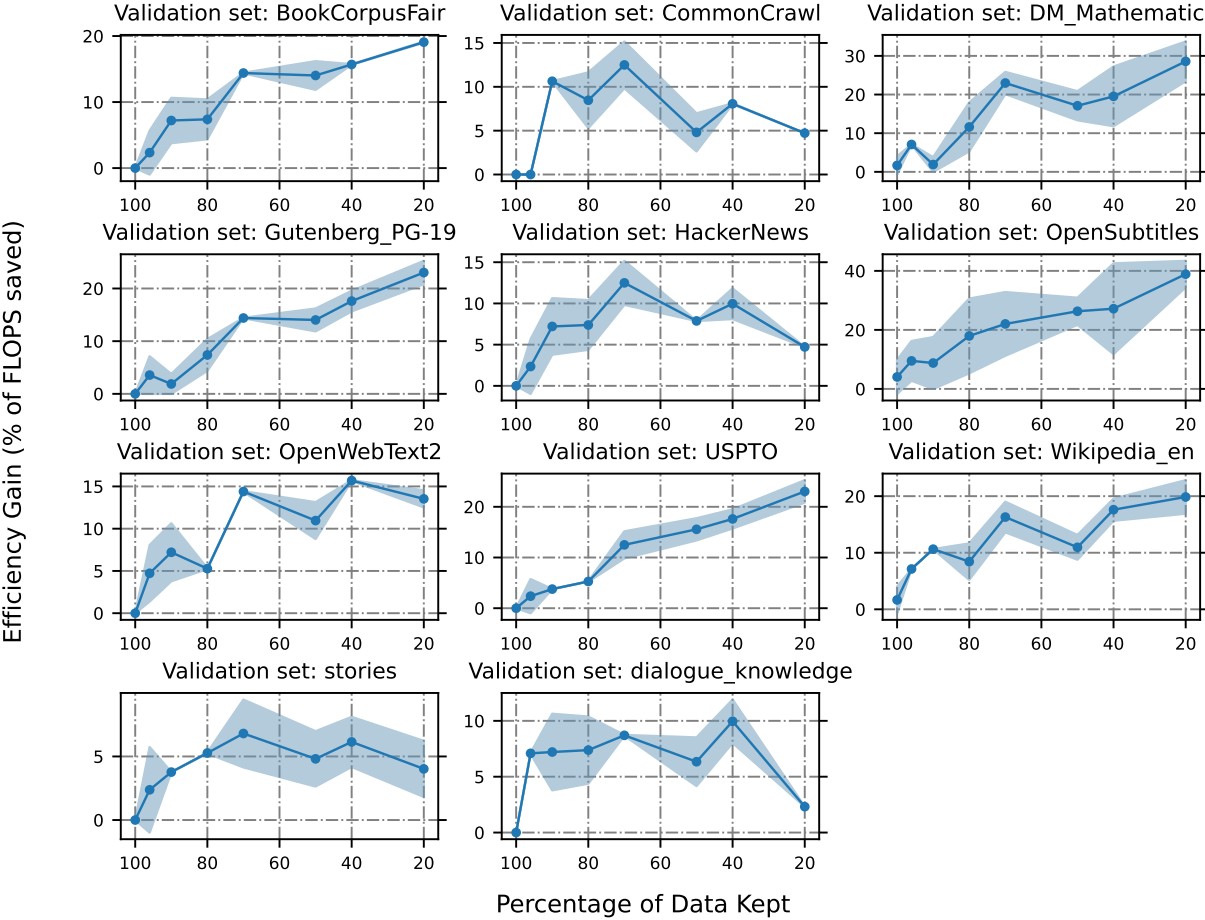

Figure A25: Percentage of Data Kept vs. Efficiency Gain on individual validation sets within opt_valid. Runs are averaged across training seeds where the model achieves baseline perplexity at some point in training, and shaded regions represent 1 standard deviation from the mean. The title of each plot represents the name of the individual validation set within opt_valid.

Keeping 90% data

| text |
| --- |
| It appears that you already have an account on this site associated with . To connect your existing account... |

Keeping 100% data (i.e. no pruning)

| text |
| --- |
| It appears that you already have an account on this site associated with. To connect your existing account... |
| You are visiting the placeholder page for Wells Williams. This page is here because someone used our placeholder... |
| You are visiting the placeholder page for Mathew Barrett. This page is here because someone used our placeholder... |
| You are visiting the placeholder page for Marcus Slatar. This page is here because someone used our placeholder... |
| You are visiting the placeholder page for Bernice Andrews. This page is here because someone used our placeholder... |
| You are visiting the placeholder page for Emiko Chille. This page is here because someone used our placeholder... |
| You are visiting the placeholder page for Landon Buckland. This page is here because Someone used our placeholder... |
| .... |
| You are visiting the placeholder page for Kylie Dickens. This page is here because someone used our placeholder utility ... |

Figure A26: Example of semantic de-duplication with SemDeDup (cluster 4500)

Keeping 20% data

| text |
| --- |
| cheap jordan shoes from china free shipping,order maroon foams , jordan blue retro 12 , jordans sz 10 , all white 14s... |
| Booming business thanks to Cristiano Ronaldo! Nike Presents Cristiano Ronaldo – CR7 Winter Collection. Cristiano Ronaldo ... |

Keeping 90% data

| text |
| --- |
| Purchase from us, you can get max discount and free shipping.Free shipping and returns on Nike Jordans at Nordstrom.com.... |
| Product range. Adidas collections are divided into three groups: Sport Performance, Originals, and Sport Style. Originals that ... |
| cool jordans for boys , foamposite paranorman , new black and white foams , lebron 1's ,cheap jordans online for sale ... |
| This Comfortable Nike Huarache Free Basketball And Running has 1600 x 900 pixel resolution with jpeg format. .. |
| Top Rating: "Best high performance product." Performance efficiency. That is the motto of our textile engineers by... |
| cheap jordan shoes online free shipping order cheap jordans for sale free shipping. Air Jordan 1's new theme color matching ... |
| ... |
| Trendy Men's Nike Kyrie 1 Best Seller 'All Star' Multicolor at high discount. Buy Nike Trainers - The Kyrie 1 All Star comes ... |

Figure A27: Example of semantically redundant de-duplication with SemDeDup (cluster 4900)

