# OpenReview forum: "SemDeDup: Data-efficient learning at web-scale through semantic deduplication"
_TMLR — Rejected by TMLR_

### Review · Reviewer_UvPc · 2023-04-17

**Summary Of Contributions:**

The authors propose SemDeDup, a straightforward embed-cluster-and-prune scheme for making giant per-training datasets smaller. Crucially, the smaller datasets can be used to train big models of similar quality, saving time/compute.

**Audience:**

Yes

**Broader Impact Concerns:**

Seems fine.

**Claims And Evidence:**

Yes

**Requested Changes:**

I am requesting these changes conditioned on your ability to run the necessary experiments. You ran so many already that I am assuming you have the compute to be able to do the few additional experiments below. If not, please let me know.

- A new column in Table 2 with a much smaller/simpler embedding [required for acceptance]
- Some sort of analysis that helps shed light on the finding in sections 5.2 and 6.4 - why would more epochs of deduped data be better than a single epoch of the original data? [required for acceptance]

**Strengths And Weaknesses:**

This is a great paper, and I have few concerns about it. I think it's ready for publication as-is, but I have two outstanding questions (see below).

Strengths
------------
- Clear writing and narrative
- Extensive experimental results
- Good results

Weaknesses
----------------
- I didn't see any mention of whether you will release your deduped training data and your codebase. This is key.
- I'd love to see a column in Table 2 that uses a much smaller model than a CLIP. Comparing two CLIPs is great, but what this doesn't tell the reader is whether embedding model has to be huge for this method to work.
- In section 5.2 you write "We observe that by training for multiple epochs over significantly pruned datasets we can reach the performance of a single-epoch run on the full dataset using 10-15% less compute." Assuming that "10%-15% less compute" means "10%-15% fewer steps at of the same batch size", this is a surprising finding. Let's consider a toy dataset which has 4 samples: x1, x2, x3, x4. And let's suppose that your algorithm finds clusters {x1, x2} and {x3, x4} from which x1 and x3 are chosen as cluster exemplars. If I understand correctly, your claim is that if my training curriculum is:

Case 1: x1, x,3, x1, x3

that this will get to the same performance 10%-15% faster than

Case 2: x1, x2, x3, x4

Right? Why would that be the case? In the extreme scenario with exact duplicates where x1 == x2 and x3 == x4, the two should be identical. In the case where x1 and x2 are semantically similar but not identical, then we have extra data diversity in Case 2 and I would expect it to be better given the wealth of results about data augmentation. I think your finding deserves more analysis and understanding - it runs contrary to a lot of previous work about the importance of data augmentation. Perhaps it's a "big data regime" finding, but that is not clear.

I think you find something similar in section 6.4: "We show that training on deduplicated LAION-440M for more iterations improves the
accuracy while still being below the number of iterations we train the baseline model for".

---

> ### Author Response · Authors · 2023-05-12
> **Response To Reviewer  "UvPc" by "SemDeDup" Authors**
>
> We greatly appreciate the questions the reviewer has raised. In the following section, we provide our responses and address the changes suggested by the reviewer.
>
> (1) Sharing the deduplicated data and SemDeDup code
>
> We plan to release our code for SemDedup in a GitHub repository, but do not include the link here so as not to break anonymity.
>
> (2) Using a smaller model for embedding the data.
>
> To further investigate the reviewer's request, we conducted additional experiments, and found that SemDeDup still works well with a smaller and simpler embedding: we were able to achieve comparable performance with a smaller model trained on ImageNet. We do note that the CLIP-Base model is a relatively small model with around 150M parameters across both the image encoder and the text encoder, though we only use the image encoder in this work to prune the data.
>
> As an additional experiment, we trained a much smaller DINO-Base model with only 85M parameters (which is 2x smaller than CLIP-Base), pre-trained on a much smaller dataset (ImageNet-1K, which is 336 times smaller than LAION).  We found that pruning using these embeddings from DINO yielded very similar performance (66.9\% for CLIP embeddings vs 66.77\% for DINO embeddings).  This shows that the performance of SemDeDup is relatively robust to significant changes to both the size of the training data and the size of the model used to find the embeddings. We have added this new result on the performance of SemDeDup using pruning based on DINO/ImageNet-1K to table 2.

---

> > ### Author Response · Authors · 2023-05-12
> > **Response To Reviewer "UvPc" by "SemDeDup" Authors**
> >
> > (3)    More explanation about why training on deduplicated data for less iterations could be better than training on raw data.
> >
> > Question asked by the reviewer:  why would more epochs of deduped data be better than a single epoch of the original data?
> >
> > We thank the reviewer for asking this insightful question. To address this question we modified Section (4.5) and the discussion section of the paper and added more elaboration about the points discussed in this response.
> >
> > Below is our response:
> >
> > We hypothesize that the reason convergence is faster on de-duplicated datasets is because the marginal information of data points is higher, which stems from the imbalance in the data — for example many images have >1000 near duplicates while some have >300,000 and few images do not have a single near duplicate (Fig. 2 in the paper). Instead of wasting compute training on the same image (or semantically similar images) from raw data, we spend compute training on data points with more marginal information.
> >
> > The case given by the reviewer
> >
> > “ Case 1: x1, x,3, x1, x3
> >
> > that this will get to the same performance 10%-15% faster than
> >
> > Case 2: x1, x2, x3, x4 “
> >
> > Is a case of a small balanced dataset (assuming that examples (x1, x2) and (x3, x4) share the same features). In practice, we found that duplicates from web-scale datasets are not uniformly distributed across examples; some data points have no duplicates, while others have several hundred thousand. This asymmetry is why training on de-duplicated data is not equivalent to simply doing more epochs on the raw data.
> >
> > Raw data has unbalanced clusters of similar examples: let’s follow a similar toy example and consider the case of (x1, x2), (x3), (x4, x5,x6), where round brackets () represent similar examples (e.g. all of them are images of the same cat with slightly different offsets). One pass through this data means that we see example x3 only once, and we see x4, x5, x6 each once, even though these three examples are very similar. It may be better to train on a dataset like ((x1), (x3), (x4)) for two epochs.
> >
> > Regarding the reviewers point about data augmentation: Note that training on raw data like ((x1, x2), (x3), (x4, x5,x6))  is different from augmenting a balanced dataset. After de-duplication, this dataset might look like ((x1), (x3), (x2)). We can then augment this dataset to get another balanced dataset ((x1, x1’), (x3,x3’), (x4,x4’)). This augmented dataset is very different from the raw data, and we agree that training on this augmented dataset could be better than training on  de-duplicated data (note that we also apply data augmentation on deduplicated data that generates more ). However, training on the raw data is most likely worse than training on the de-duplicated dataset because of imbalance.
> >
> > We also note that historically, manual cleaning of datasets has improved training convergence — even datasets like ImageNet were cleaned manually to some extent and it was shown this improved training convergence. However, with large-scale datasets it becomes difficult to do manual filtering; in this work, we provide a method to automatically clean large billion-scale datasets.
> >
> > Another important usage of data duplication methods like SemDeDup that goes beyond beating training on the whole data is high-quality subset selection.
> >
> > High-quality subset selection: Imagine we are trying to train models that do not need to train on the entire dataset, or we are academic researchers limited by compute. In this case, it would be beneficial to have access to high-quality subsets of the large datasets. For example, given a large dataset like LAION (that exceeds 5 billion examples), we can select a small subset (e.g. 5\% of the dataset) to train a model that requires less compute. In this case, our goal is not to make the 5\% dataset better than the whole dataset for training. Instead, the core goal is to find a relatively small subset (5\% for example) that does not contain exact, near, and semantic duplicates.
> > We modified Section (4.5) and the discussion section of the paper and added more elaboration about the points discussed in this response.

---

> > > ### Comment · Reviewer_UvPc · 2023-05-12
> > >
> > > Thank you for the thorough responses. I feel that my questions have been well-answered.

---

### Review · Reviewer_sFqx · 2023-04-20

**Summary Of Contributions:**

The authors proposed an algorithm, SemDeDup, to remove near duplicates from the training data of large DNN models. Firstly, the algorithm maps each data sample into an embedding space that captures the semantic relationships. The mapping is done by pre-trained models on similarly distributed data. Then, K-means clustering is conducted in the embedding space with a predetermined number of clusters (k). A full pairwise similarity is computed within each cluster, and a threshold is used to further segment a k-means cluster into disjoint subsets of near duplicates. Only one of the data samples is kept as the new training data set. Varying the threshold allows a continuous control of how much data is left.

The authors tested the algorithm on two different data sets, LAION (images) and C4 (text). Evaluation shows that SemDeDup can remove up to 50% of LAION data without hurting the prediction accuracy. This directly translates to a significant save of training time and thus cost. The reduction is smaller on C4 (15%) given that it's a more curated data set and some duplicates may have been removed already.

The authors discussed the computational viability of the algorithm. Although the k-means step doesn't really reduce the complexity resulted from the all pairwise similarity (at O(n^2)), it does reduce the practical computation time by 5x on a typical web-scale data set like LAION. This makes the algorithm attractive for practical use.

The authors also studied the sensitivity of the choice of hyperparameters, including number of clusters (k), similarity threshold, pre-trained models. It also evaluated different strategies of choosing the representative sample from a near-dup cluster. The algorithm is shown to be quite stable under different choices of hyperparameters.

Lastly, the authors discussed the limits of the algorithm in the possibility of better deduping methods, reliance on pretrained models and relatively low gain in C4 compared with LAION.


**Audience:**

Yes

**Broader Impact Concerns:**

No concern.

**Claims And Evidence:**

Yes

**Requested Changes:**

Color legend in Figure 1 (a). The red/blue removed examples are supposed to be in lighter color but not.

We can be upfront to say that the way to pick a sample from a near-dup cluster doesn't matter at the beginning of Section 3, instead of saying we need to pick the lowest cosine similarity to the centroid and let the readers figure it out later in the paper.

Page 6 first paragraph: " This results in training for a fewer number of iterations when training on
deduplicated data, thereby achieving efficiency gains.". This sentence implies that reduced iterations is the reason of the efficiency gains, while I think it's related to the reduction of training data, not necessarily the reduction of # iterations. Am I getting this wrong?

The paper makes frequent references to the Appendix which is quite sizable. For example in Section 4.3. Please consider moving important data/figures into the paper.

Section 6.1 line 2: "section 3" should be "Section 3". Minor issue. Seen elsewhere in the paper too.

Figure 7: It seems that Lee et al (2021) is quite effective at getting the same level of perplexity, while having significantly lower computational complexity.
Section 7 first sentence: "SemDeup" -> "SemDeDup".


**Strengths And Weaknesses:**

Strength:

1. Thorough experiments. The authors conducted extensive experiments on large scale datasets to test different aspect of the model. For example across images and text, and on all choices of hyperparameters.

2. Good explanation on intuition. The authors gave many examples of semantic duplicates, giving readers an intuitive knowledge about what data are removed from the dataset.

3. Discussion on limitations. The authors gave a candid discussion on the limits of SemDeDup in the last section.

Weakness

1. Weak baselines. Random dropping is a pretty weak baseline. There are many image clustering algorithms that don't require expensive pairwise comparison. SemDeDup should compare with them to help understand the balance between quality and computational cost.

2. The paper called it semantic duplication removal, while most of the examples given are really just direct duplication (different clipping etc). It would be interesting to see up to which point of similarity that the model quality started to decay. Is near-dup the line to draw for input data cleansing?

---

> ### Author Response · Authors · 2023-05-12
>
> We deeply appreciate the thoughtful questions raised by the reviewer. In the subsequent section, we will provide comprehensive responses and address the suggested changes proposed by the reviewer.
>
> 1) Color legend in Figure 1 (a). The red/blue removed examples are supposed to be in lighter color but not.
>
> Great catch! We have corrected this in the updated paper.
>
> 2) Our response to: “We can be upfront to say that the way to pick a sample from a near-dup cluster doesn't matter at the beginning of Section 3, instead of saying we need to pick the lowest cosine similarity to the centroid and let the readers figure it out later in the paper.”
>
> We greatly appreciate the reviewer's attention to this crucial detail. To ensure clarity for readers, we have made slight modifications to the method description section in the paper. Thank you for bringing this to our attention.
>
> 3) Our response to: “ Weak baselines. Random dropping is a pretty weak baseline”
>
> We studied the relationship between other baselines like hashing methods and SemDeDup. Our evaluation, using Difference Hashing (dhash), as an example, indicates that these methods are unlikely to perform as effectively as SemDeDup in detecting semantic duplicates. While we did not run experiments on these methods due to the computational cost (each individual CLIP training requires ~12k GPU hours), our findings suggest that while hashing methods may be good at detecting exact duplicates, they perform weakly in identifying semantic duplicates.
>
> Why dhash is not a good metric for semantic image similarity in contrast to cosine similarity?
>  - We found that the intersection between examples removed by dhash and cos similarity is very small on the LAION-440M dataset.
>  - For diverse web data, there is no strong correlation between dhash and cos similarity for detecting duplicates. For example, on LAION-440M we measured a Pearson's correlation of 0.051 excluding exact duplicates from our correlation measurement.
>
> We supported this study with a plot for the correlation and visualizations and added them to Fig. A4 in the paper.
>
> 4) Our response to: “The paper called it semantic duplication removal, while most of the examples given are really just direct duplication.”
>
> To some extent, this depends on how you define semantic duplicates. We define semantic duplicate as two data points which are not identical and would not be detected by exact de-duplications or minhash-based techniques, but which are derived from the same data point. For example, many of the semantic duplicates we find are derived from the same source image, but may have been distorted by blur, change in aspect ratio, resolution, padding, etc. We would therefore argue that most of the examples we discover are not merely the result of “direct duplication.”
>
> We also do not consider our method as a method for removing near-duplicates only, but can also remove conceptually similar, but distinct examples (what we call “semantic redundancy”). Instead, near-duplicates are removed at a low deduplication threshold value, and we provide a smooth way to reduce the dataset size to any size we want by changing the deduplication threshold value $\epsilon$. Increasing  $\epsilon$ will not only remove near-duplicates, but will also remove all examples with that level of semantic similarity defined by  $\epsilon$. For example we show in Fig. 4, 7, and A22 examples of removing 80\% of the data and training on only 20\% of it. Similarly, the method can be used to deduplicate the dataset to any size smaller than 20\%. At these extreme pruning percentages, the majority of removed data points are not derived from the same source data but rather are conceptually similar.
>
> We updated Fig. A12 and A13 to show more examples from the LAION-440M dataset after removing 70\%, and 80\% of the data examples.
>
> 5) Our response to: " This results in training for a fewer number of iterations when training on deduplicated data, thereby achieving efficiency gains. This sentence implies that reduced iterations is the reason of the efficiency gains, while I think it's related to the reduction of training data, not necessarily the reduction of # iterations. Am I getting this wrong?”
>
> We define “efficiency gain” as using less compute to reach baseline model performance. For example, if someone trains a model on a smaller dataset but for more total iterations, then we would say that the training efficiency is lower. Also, note that we use the same number of GPUs, batch size, and learning rate schedule for training all models. We keep these factors fixed so that we can fairly compare training efficiency of different models we train. Since the batch size is constant, reducing the amount of data directly leads to a reduction in iterations and a reduction in total training time and compute.

---

> > ### Author Response · Authors · 2023-05-12
> >
> > 6) Our response to: "It seems that Lee et al (2021) is quite effective at getting the same level of perplexity, while having significantly lower computational complexity.”.
> >
> > First and foremost, we note that the real cost of SemDeDup is small compared to the cost of training. We compute these figures exactly in Tables A3 and A4, showing that the cost of SemDeDup on LAION costs less than a third of an epoch of CLIP training. We also emphasize that SemDeDup only needs to be performed once at data construction time. For these reasons, we argue that the computational complexity of SemDeDup is reasonable for very large web-scale datasets; if one has enough compute to train on a web-scale dataset, one has enough compute to perform SemDeDup.
> >
> > However, we also disagree with the assessment that our method has worse computational complexity than Lee et al., 2021:
> >
> > Let $N$ denote the total number of documents in the dataset. The method from Lee et al (2021) — i.e. the NearDup method we compare to in our paper — has three conceptual steps:
> >
> > (A1) Tokenize and estimate the Jaccard index between documents via minhash. Document pairs are considered “potential matches” if the hashes for a document fall into the same bucket. This step is $O(N$).
> >
> > (A2) For all potential matches, calculate the * actual * Jaccard index [see Appendix A, 1st paragraphs after the equation]. If all documents fall into the same bucket, this is worst case $O(N^2)$.
> >
> > (A3) For all potential matches with actual Jaccard index above 0.8, compute the edit similarity. This is done just sequentially computing edit similarity — they construct a graph with $N$ nodes (one for each document, and an edge between documents if their edit similarity is above 0.8), and then find connected components in this graph, which takes $O(N)$ time since the graph has $O(N)$ nodes and edges. However, note that this is an approximation to the edit similarity.
> >
> > For SemDedup:
> >
> > (B1) We cluster the documents with k-means which takes $O(N)$ .
> >
> > (B2) For each cluster, we compute pairwise cosine similarity, which takes worst case $O(N^2)$ time, if all documents fall into the same bucket.
> >
> > In other words, the computational cost of minhash depends on how well balanced the buckets are; similarly, the computational cost of SemDedup depends on how well balanced the clusters are.
> >
> > Also note we can easily remove the $O(N^2)$ factor in SemDedup by approximating the pairwise cosine similarity within clusters, similar to how Lee et. al approximates the edit similarity within buckets. We chose not to do so, in order to explore the “exact” version of SemDedup.
> >
> > Outside of the computational complexity, we note that methods like MinHash do de-duplication on a particular input space. If we have multiple input spaces (e.g. multimodal models), we would need to run separate hashing methods in each modality, and figure out how to combine the results from each modality. With SemDeDup, we can run de-duplication once in the joint embedding space, thereby reducing the actual amount of compute needed to perform de-duplication.

---

### Review · Reviewer_Liyc · 2023-04-27

**Summary Of Contributions:**

Proposes to de-duplicate unlabeled image or text datasets by removing near-duplicates in the embedding space of a pretrained model. The paper provides a simple method for performing deduplication and reports on an experimental study on LAION and C4 datasets. The paper provides further evidence that removing near-duplicates can be beneficial and, as such, makes a useful contribution. The paper is not ready, however, as the underlying idea is folkore, the proposed method not novel, and the experimental study not sufficiently insightful.


**Audience:**

Yes

**Broader Impact Concerns:**

None.

**Claims And Evidence:**

Yes

**Requested Changes:**

Critical: Tone down language and discuss related work on points 1 and 2. Expand/rewrite experimental study to address point 3.

Uncritical: The paper heavily refers to figures in the appendix, which makes reading cumbersome.


**Strengths And Weaknesses:**

I discuss the strength and weaknesses of individual aspects separately.

1. Deduplication in embedding space

Strength: A good idea!

Weakness: Well-known. -> It's a folklore approach, there are many blog posts about it, it's a part of the deduplication approach proposed by Sorscher et al. (2022), there are libraries doing this for deduplication (e.g., imagededup, fastdup, snip-dedup), and it's also used for related problems such as finding easy and hard examples.

2. Method for deduplication

Strength: Sufficiently fast, gives useful results.

Weakness: A well-studied problem. -> First, the approach used in the paper is to apply k-means on the embedded examples, following Sorcher et al. (2022). The paper differs in how to select examples within each cluster, but it's not clear to what the impact of this difference actually is (over, e.g., using the most central or most far-away points per cluster as Sorcher et al.). Second, the purpose of k-means in this approach is to perform "blocking", a problem that has been studied e.g. for entity linking. A go-to approach is LSH. Third, the approach is related to finding an maximal independent set on the epsilon-neighborhood graph and, in particular, the greedy method to find such a set. It's applied per cluster here. Finally, it's not clear whether blocking is actually needed or whether decent similarity search is enough (also a well-studied problem with many available libraries).

3. Experimental study.

Strength: The paper reports on the kind of duplicates found in (different subsets of) LAION as well as C4, highlighting that the datasets suffer from near-duplicates. It also performs a study that sheds some light into the effect of removing these duplication on model performances. Finally, I assume that the authors publish the reduced datasets, a potentially useful resource. To me, this study is the actual contribution of this paper.

Weakness: The study is confusing and it's hard to take insights from it.

The key metrics one may be interested in is (i) model performance as a function of compute cost and (ii) best achievable model performance. The paper seems to focus on (i), but it is is often not clear what the compute cost actually is. This is because most experiments modifies dataset size and training cost simultaneously. This makes the results hard to digest and partly also misleading (e.g., Fig 4). The paper does not discuss (ii) at all. It should present SOTA results for the used benchmarks and discuss to what extent these results can also be obtained on de-duplicated data. This is important because the datasets for which this approach is relevant are datasets used for pre-training and the main goals are to either improve performance of the pre-trained model or to reduce the cost for the same performance.

Weakness: Alternative methods are not discussed (with one exception).

Alternative deduplication methods, such as the ones mentioned under (1) above as well as the ones cited in the paper, are not studied at all. (Also, many of these approaches are barely described, if at all.)

---

> ### Author Response · Authors · 2023-05-12
> **Response to Reviewer Liyc (part 1)**
>
> We thank the reviewer for their thoughtful questions. In the following section, we will thoroughly respond to these inquiries and carefully address the proposed changes.
>
> 1) Tone down language and discuss related work on points 1 and 2.
>
> We thank the reviewer for pointing out all these related works. We have added the proper citations/descriptions to Section (2) of our paper under “Methods for de-duplication”.
>
> 2) Weakness 1:  Comment about  “.... the underlying idea is folklore, the proposed method not novel...”
>
> We strongly believe that the power of our method lies in its simplicity. Methods like cosine similarity for measuring data point similarity have been widely utilized, and have their pros and cons. However, we are not inclined towards complex methods when simpler approaches work highly effectively. SemDeDup, as a deduplication technique, offers ease of implementation and can be extended to various modalities beyond just images and text.
>
> For example, this week, an independent startup, Mosaic ML, announced that they utilized SemDeDup to de-duplicate their training data, and trained extremely large language models on it. The immediate implementation and execution they achieved would not have been possible without the simplicity of the method!.
>
> While we appreciate that the reviewer agrees that deduplication in the embedding space is a good idea, we do not believe the fact that it is a simple idea should be taken as a weakness of our paper, but rather a strength, especially given the strong efficacy of our approach.  Also, while prior methods exist to do deduplication in various spaces, to our knowledge, nobody has applied this idea to web-scale data and shown that one can reduce the size of such web-scale data by 50\% (e.g. LAION) without suffering substantial accuracy loss.  Furthermore, nobody has shown that one can reduce the size of NLP datasets like C4 and train for longer, achieving the same accuracy with 10-15\% less compute. In fact, the closest published work to ours, Lee et al., 2021, performs deduplication in an n-gram embedding space with MinHash, and only removes 4\% of the data from C4. We would argue that the primary source of our better results is the use of a pre-trained embedding space and a looser search procedure than MinHash. These are our main contributions, and we will release both the smaller, higher quality datasets and the code; we believe both of these will have a substantial impact on the field of large foundation model training.
>
> We also note that in the TMLR acceptance criteria web page mentions that, "novelty" is explicitly not considered a primary criterion for acceptance.
>
> Quote from TMLR Acceptance Criteria.
>
>      “... Crucially, it should not be used as a reason to reject work that isn't considered “significant” or “impactful” because it isn't achieving a new state-of-the-art on some benchmark. Nor should it form the basis for rejecting work on a method considered not “novel enough”, as novelty of the studied method is not a necessary criteria for acceptance. We explicitly avoid these terms (“significant”, “impactful”, “novel”), and focus instead on the notion of “interest”...”
>
> 3) Weakness  2: Alternative deduplication methods, such as the ones mentioned under (1) above as well as the ones cited in the paper, are not studied at all. (Also, many of these approaches are barely described, if at all.)
>
> We studied the relationship between other baselines like hashing methods and SemDeDup. Our evaluation, using Difference Hashing (dhash), as an example, indicates that these methods are unlikely to perform as effectively as SemDeDup in detecting semantic duplicates. While we did not run experiments on these methods due to the computational cost (each individual CLIP training requires ~12k GPU hours), our findings suggest that while hashing methods may be good at detecting exact duplicates, they perform weakly in identifying semantic duplicates.
>
> Why dhash is not a good metric for semantic image similarity in contrast to cosine similarity:
>
> - We found that the intersection between examples removed by dhash and cos similarity is very small on the LAION-440M dataset.
> - For diverse web data, there is no strong correlation between dhash and cos similarity for detecting duplicates. For example, on LAION-440M we measured a Pearson's correlation of 0.051 excluding exact duplicates from our correlation measurement.
>
> We supported this study with a plot for the correlation and visualizations and added them to Fig. A4 in the paper.

---

> > ### Author Response · Authors · 2023-05-12
> > **Response to Reviewer Liyc (part 2)**
> >
> > 4) Expand/rewrite experimental study to address point3 (best achievable model performance.)
> >
> > We applied two groups of experiments in the paper 1) training for the same number of iterations as the baseline model, hence for fewer iterations on deduplicated datasets. 2) Continuing training for longer on deduplicated data to achieve better performance. We note that these are not mutually exclusive and one can smoothly trade-off efficiency for performance.
> >
> > To make it clear for readers we discuss the results of (1) and (2) in two different sections in the main text of the paper (Section 4.4 for (1) and Section 4.6 for (2)).
> >
> > We show results of (1) in Fig. (4), (5), (6), and (7). These results focus on how efficient learning from deduplicated data is.
> >
> > Regarding (2) we conducted another set of experiments to see what is the best performance we can reach if we train longer, considering that we train for fewer iterations in Fig. 4, 5, and 7. We discuss that in Section (4.6) and show results in Tables A1 and A8. For example, we show that the best zero-shot performance on ImageNet for training on 50\% of the data is 68.92\% when training for 75\% of the number of iterations. In Table A8 we show the results of training on deduplicated data of sizes  40\%, 50\%, 60\%, and 70\% for a number of iterations that match the baseline model training.
> >
> > For training the CLIP model on whole data we follow the original CLIP paper (Radford et. al 2021)  and train for 32 epochs. (Radford et. al 2021)  trained CLIP for 32 epochs on 400M examples, to make sure that the training is complete we train on 440M examples for 32 epochs also (instead of 30 epochs which match the number of training iterations) until the performance starts to saturate. This means that we train for more iterations than (Radford et. al 2021).
> >
> > 5) "It's not clear whether blocking is actually needed or whether a decent similarity search is enough."
> >
> > Why clustering is important?
> >
> > - Clustering can reduce the search complexity of de-duplication from $O(n^2)$ to $O((\frac{n^2}{k}))$. Where n is the dataset size and k is the number of clusters. For example, the average cluster size for the LAION-440M dataset when $k$=50000 is 8,748 which is 50,000 times smaller than the dataset size.
> > - In Appendix A.2, we show for different values of $k$ that, while clustering reduces complexity, it is also very effective in detecting most of the duplicates in the dataset.
> >
> >
> > 5) Finally, we would like to add a few comments on the method:
> >
> >  i) Comment on the similarities and differences between SemDeDup and the method in Sorscher et al. (2022)
> >
> > We would like to point out a couple of differences between our work and some of the related work that the reviewer mentioned.
> > While the reviewer suggests our work is very similar to Sorscher et. al. 2022 (their self-supervised (SSL) prototypes method), we note that there are major differences between SemDeDup and Sorscher et al. (2022). SSL Prototypes aims to remove prototypical examples within each cluster; SemDeDup aims to remove semantic duplicates, regardless of the broader cluster structure. Indeed, before embarking on this work, we applied Sorscher et al. (2022), on the LAION dataset and found that, while it works on ImageNet, it fails to outperform random pruning the LAION dataset. We hypothesized that the reason is that ImageNet is a cleaned and balanced dataset while LAION is an unbalanced web dataset, potentially with many highly redundant and semantically similar images.  Thus we departed significantly from Sorscher et. al. (2022), which clustered the ImageNet dataset into around 1000 clusters (corresponding to 1000 classes) and kept data points furthest from the cluster centroid. We instead aimed to keep only one data point amongst any group of data points with high similarity.  For us, the step of clustering (e.g. finding 50,000 clusters for LAION) was only an approximation step that enabled us to avoid computing pairwise similarity between all data points in LAION.  We also further did not require any class balancing that was important Sorscher et. al. (2022).
> >
> >
> >    ii)  Note about:  “...Proposes to de-duplicate unlabeled image or text datasets by removing near-duplicates in the embedding space of a pretrained model…”
> >
> > We do not consider our method as a method for removing near-duplicates only. For example, when we reduce the dataset down to below 40\% or so, many data points that are removed are not near-duplicates of the remaining data points; instead, they have some kind of loose semantic similarity but are clearly distinguishable as distinct images by humans.  This is a strength of deduplication in the embedding space. A further strength of deduplication in the embedding space is that it allows us to apply the same steps regardless of the modalities (images, text, videos, audio, etc) the embeddings are computed from. For example, we apply the same steps and the same code on LAION image embeddings and C4 document embeddings.

---

> > > ### Comment · Reviewer_Liyc · 2023-05-24
> > > **Thoughts on response/revision**
> > >
> > > I'd like to thank the authors for their detailed response. My key points were:
> > >
> > > > Tone down language w.r.t. novelty and discuss related work on points 1 and 2. Expand/rewrite experimental study to address point 3.
> > >
> > > These points have not been adequately addressed. In fact, the paper claims (i) a novel method, (ii) a simple method, (iii) that this new method outperforms prior work, and (iv) that the method is able to achieve SOTA performance with less cost. The first three points are not sufficiently argued for in the paper, the last one needs a bit more context.
> > >
> > > On (i). The authors do acknowledge that pruning in embedding space is not a new idea in their response. The method is essentially an approach for coreset selection and, in particular, a geometry-based method (see, e.g., the discussion in Guo et al. (2022)); this connection is not acknowledged in the paper. In contrast to prior work in the coreset selection area, however, the method is probably more scalable. That being said, the scalability comes from the ideas of Sorcher et al. (2022), as the authors also acknowledge in their response. All of this is generally OK for me/TMLR, but it would need to be discussed more clearly (and right from the beginning). Currently, the paper claims more novelty than there is.
> > >
> > > On (ii). Next, the authors argue for the simplicity of their method. My point was that even simpler approaches may work just as well: e.g., directly use a fast out-of-the-box cosine similarity search method to obtain (and then remove) duplicates within the similarity threshold. It not clear that clustering followed by all-pairs similarity computations within a cluster would outperform such a method w.r.t. to scalability or quality. In fact, the paper uses k-means clustering, which needs to perform O(Nk) similarity computations per round, where k is the number of clusters. Consequently, clustering complexity is not O(N) as claimed, but O(Nk). And k is quite large in the specific setting (up to 50000). So, yes the proposed method is not very complex, but no, it's not the simplest method that comes to mind. Again, and as stated in my original review, all of this would be OK with me if the paper discussed in adequately. It does not, however.
> > >
> > > On (iii). The paper claims that their method is superior to prior deduplication methods. First, the paper does compare to random pruning, but this is the weakest possible baseline. It also compares to Lee et al. (2021), which it does outperform and which does constitute a useful insight. It does not, however, compare to Sorcher et al. (2022) and libraries such as fastdup. These comparison would be needed, however, to claim superiority w.r.t. to the state-of-the art.
> > >
> > > On (iv). The paper now shows that their method outperforms full-data training on Laion440M, which is an insightful addition to the paper and should go to the main text. My main gripe is that the obtained accuracy (69%) is not related to what's possible on Laion440M (e.g., OpenCLIP reports 72.77% with ViT-L/14). This point would be easy to fix, e.g., by stating/arguing for it or (preferably) by adding results with ViT-L/14.
> > >
> > > I'd also like to reiterate my point that the experimental study is hard to read as "it is is often not clear what the compute cost actually is". The standard way to report results is to consider the number of training samples being used (e.g., 440M with 10 epochs = 220M with 20 epochs = 4.4B data points). Many of the results would be much more interpretable if this quantity were shown on the x axis.
> > >
> > > In summary: To me, the main contribution of the paper is to provide further evidence that embedding-space pruning can be beneficial. This is a valid and interesting contribution. The paper is not positioned in this way, however, but instead claims (i) a novel method , (ii) a simple method), and (iii) outperforming prior methods. These three points lack justification, however.

---

> > > > ### Comment · Reviewer_Liyc · 2023-05-24
> > > > **On pre-trained models**
> > > >
> > > > One additional point: the method proposed here uses a "pre-trained CLIP model" to remove duplicates in many experiments, and then retrains a CLIP model on the results. But if one has a pre-trained CLIP model, why retrain it? An exception is Section 6.2, which does provide some insight in this direction by using other base models. Generally, the authors should argue more clearly here, state which pretrained model was used right away in Sec 4.1, and relate the results to the performance of the baselines models.

---

> > > > > ### Author Response · Authors · 2023-05-31
> > > > > **Response to Reviewer "Liyc": Part 1**
> > > > >
> > > > > Thank you for your response and feedback. We address the concerns raised by the reviewer in the points below:
> > > > >
> > > > > (i) Novelty:
> > > > >
> > > > > We appreciate the reviewers’ concern regarding novelty and over-claiming, and want to be sure that we do not mislead the reader with respect to the originality of our approach. However, we respectfully disagree with the reviewers’ contention that we claim that our method is novel in the paper. We want to emphasize that we did NOT use any words or phrases that claim novelty (novel, new, etc.) to describe SemDeDup in our paper. If there are particular sentences the reviewer feels are misleading, please draw our attention to these specific sections and we are happy to change them.
> > > > >
> > > > > Regarding discussing the use of embeddings, here is a quote from the Related Work Section on page 4 under “Methods for deduplication” from the current version of the paper:
> > > > >
> > > > > “….Another category of techniques involves utilizing embeddings from pre-trained models to represent the data. The idea of working in the embedding space has been utilized in many works for coreset selection (Sinha et al., 2020), (Sener & Savarese, 2017) (Agarwal et al., 2020….”.
> > > > >
> > > > > Also here is another quote from the Related Work Section on page 4 under “Methods for deduplication”:
> > > > >
> > > > > “…(Webster et al., 2023), and Jain et al. (2019) apply deduplication in the embedding space on different scales but without providing a strong study of training on deduplicated data…”
> > > > >
> > > > >
> > > > > Regarding linking our work to coreset selection and the work in Guo et al., 2022 , see the Related work section on page 4:
> > > > >
> > > > > “....Beyond deduplication, a host of classical machine learning approaches seek to achieve data efficiency by finding coresets, defined as small subsets of the training data that can be used to train a machine learning algorithm to the same test accuracy achievable when training on the entire training data (see e.g. (Guoet al., 2022; Phillips, 2016) for reviews)....”
> > > > >
> > > > > (ii) Using Clustering in the embedding space:
> > > > >
> > > > > Regarding using clustering see the Related Work Section on page 4 where we acknowledge that we are not the first work to use this idea. However, we wish to emphasize that the way Sorscher et al., 2022 used the clusters to determine data to remove is fundamentally different from our approach. Sorscher et al., 2022’s method, SSL Prototypes, uses clustering to measure “prototypicality”, and removes data points which are close to the cluster centroid. In contrast, SemDeDup aims to remove points which are highly similar to one another, regardless of their location relative to a cluster centroid. As a result, the data each method removes is very different. Because SSL Prototypes removes data near centroids, it will remove data whether or not it has a close neighbor, so long as it’s near the centroid. Relatedly, SSL Prototypes would not remove two data points that are duplicates if they are far away from the centroid – in fact, SSL Prototypes would effectively enrich these duplicated data.
> > > > >
> > > > > To better illustrate the differences between SemDeDup and SSL Prototypes, we added a figure (Fig. A5) comparing the two methods and illustrating why the method by Sorscher et al. (2022) might fail in detecting duplicates since it only considers the distance to cluster centroid and does not consider the semantic similarity between duplicates.
> > > > >
> > > > > We’ve also modified the section in the Related Work discussing Sorscher et al. to make these differences as clear as possible:
> > > > >
> > > > > “…(Sorscher et al., 2022) proposed using clustering in the embedding space of a pre-trained foundation model to rank examples based on their usefulness for training. Their method aims to remove prototypical examples within each cluster by ranking examples without considering their semantic similarity to each other, but rather by comparing each to the cluster centroid and removing those points which are closest to the centroid. As a result, pairs of duplicates are assigned similar ranks, and they are either kept or removed together. In this work, we also utilize clustering, but we instead aim to remove semantic duplicates by focusing on nearby points regardless of their location. In addition, we move from relatively small, highly curated ImageNet scale to highly uncurated, web-scale data.
> > > > > ….”

---

> > > > > > ### Author Response · Authors · 2023-05-31
> > > > > > **Second Response to Reviewer "Liyc": Part 2**
> > > > > >
> > > > > > (iii) Simplicity
> > > > > >
> > > > > > ”
> > > > > > Naively searching for duplicates has a time complexity of $O(n^2)$ where $n$ is the number of data points. Comparing all examples in a dataset to each other can find all duplicates, but this approach is impractical for large web-scale data.  For example, the LAION-440M dataset would require $\approx 1.9 \mathrm{x} 10^{17}$ similarity computations. Our primary objective is therefore to reduce the computational cost of these comparisons. The k-means clustering step in SemDeDup reduces the complexity of searching for duplicates substantially from $O(n^2)$ to $O(\frac{n^2}{k})$ assuming approximately uniform cluster size\footnote{Note that our choice of $k$ depends on $n$, it is not a constant in the context of this complexity analysis.}. This means we only require $\approx 3.9 \mathrm{x} 10^{12}$ intra-cluster comparisons instead of $\approx 1.9 \mathrm{x} 10^{17}$ across all pairs, a 5-order of magnitude improvement.
> > > > > >
> > > > > > Considering that k-means clustering has a time complexity of $O(k.n)$ (See Appendix A.1 for more details). The time complexity of SemDeDup is upper bounded by the clustering step complexity ($O(kn)$) or by the search step complexity ($O(\frac{n^2}{k})$). The search step dominates when $\frac{n^2}{k}> kn$ or simply when $k<\sqrt{n}$ and the complexity of SemDeDup, in this case, is $O(\frac{n^2}{k})$. While the clustering step dominates when $k\ge\sqrt{n}$ resulting in a time complexity of $O(k.n)$.
> > > > > >
> > > > > > As long as $1<k<n$ the time complexity of SemDeDup ($O(\frac{n^2}{k})$ or $O(kn)$) stays below $O(n^2)$. In numerical terms, this corresponds to $\approx 3.9 \mathrm{x} 10^{12}$ or $\approx 2.2 \mathrm{x} 10^{13}$ pairwise comparisons respectively, for LAION-440M dataset when $k=50,000$, which is 5 or 4 orders of magnitude faster than conducting $n^2$ ($\approx 1.9 \mathrm{x} 10^{17}$) comparisons.
> > > > > > “
> > > > > >
> > > > > > While we don’t claim that our method is the ”simplest” method, we believe that the simplicity of a method does not include only the implementation, but also includes time complexity. For example, while naive search is simple to implement, Its time complexity is high enough to be impractical for large-scale datasets where n could be several billions of examples ($O(n^2)$).
> > > > > >
> > > > > > We agree with the reviewer that the time complexity of clustering is not $O(n)$. Actually, we didn’t state in the paper that it is $O(n)$.  We added a completed section to Appendix A.1 discussing the
> > > > > > time complexity of clustering. When comparing the time complexity of $O(ikn)$ to $O(n^2)$ we have to recall that $k$ << $n$ for large-scale datasets. For example, we use $k$ = 50,000 while $n$ is 440,000,000 for the LAION-440M dataset.

---

> > > > > > > ### Author Response · Authors · 2023-05-31
> > > > > > > **Second Response to Reviewer "Liyc": Part 3**
> > > > > > >
> > > > > > >
> > > > > > > (iv) Other baselines:
> > > > > > >
> > > > > > > We added two additional baselines using the methods in Sorcher et al. (2022) and using a hashing algorithm.
> > > > > > >
> > > > > > > We implemented the method in Sorcher et al. (2022) and applied self-supervised prototypes (SSP) pruning with cluster balancing on LAION-440M dataset. We pruned the dataset to
> > > > > > > to 40\% of it is size and report the following results for zero-shot top1 accuracy on ImageNet:
> > > > > > >
> > > > > > > -SemDeDup:                                   imagenet-zeroshot-top1: 66.90 \%
> > > > > > >
> > > > > > > -Random:                                        imagenet-zeroshot-top1: 64.80\%
> > > > > > >
> > > > > > > -SSP pruning w/ cluster balancing: imagenet-zeroshot-top1: 61.89\%
> > > > > > >
> > > > > > > We also implemented another baseline using a hashing algorithm as in the (imagededup, and imagehash libraries):
> > > > > > > To avoid doing $n^2$ comparisons we used clustering in the embedding space, and then we used Hamming distance as a metric. In this case, we replace computing dot product (cosine similarity) in SemDeDup with computing the Hamming distance between hash codes. We keep the clustering step to reduce the number of Hamming distance comparisons between examples. To hash the images, we used Difference Hash (dhash) with hash codes of length 256.
> > > > > > > To study the efficiency of this method, we trained two new baseline models using 42% and 63\% of LAION-400M, respectively. We found that SemDeDup with cosine similarity outperforms both models when using 40\% and 63\% of LAION-440M, respectively.
> > > > > > >
> > > > > > > Results after training on 63\% of LAION-440M for 32 epochs (8.8B examples seen). See Table A3:
> > > > > > >
> > > > > > > -SemDeDup (63\%):              imagenet-zeroshot-top1: 68.62\%
> > > > > > >
> > > > > > > -Clustering+Dhash (63\%):    imagenet-zeroshot-top1: 67.24\%
> > > > > > >
> > > > > > > Results after training on 40\%  (or 42\%) of LAION-440M for 32 epochs (5.6B examples seen), See Table A3:
> > > > > > >
> > > > > > > -SemDeDup (40\%):              imagenet-zeroshot-top1: 66.90\%
> > > > > > >
> > > > > > > -Clustering+Dhash (42\%):   imagenet-zeroshot-top1: 64.10\%
> > > > > > >
> > > > > > > -Random Pruning (40\%):     imagenet-zeroshot--top1: 64.80\%
> > > > > > >
> > > > > > > For dhash, we resize all images to a resolution of 16x16 and then generate hash codes from them (the time complexity of this step is $O(w * h)$, where $w$ and $h$ represent the width and height of the image, respectively).  Data size is 42\% with dhash because Hamming distance is a discrete function. Hence, we are restricted to limited values of deduplication thresholds ($\epsilon$) to use, so achieving the exact size of 40\% is not possible with any of the available $\epsilon$ values.
> > > > > > >
> > > > > > > In summary, we show that hamming distance is not as good as cos distance in the embedding space in measuring semantic similarity between images.
> > > > > > >
> > > > > > > v) Performance comparison to OpenCLIP
> > > > > > >
> > > > > > > We emphasize that in all of our experiments, we trained a ViT-B-16 model. The most relevant comparison in the OpenCLIP repo is therefore 67.1\% (ViT-B-16 trained on LAION400M) as compared to our results of 69\%. Our model is therefore outperforming the most comparable model in OpenCLIP – not underperforming as suggested. We hypothesize that the increase in performance we observe comes from our using LAION440M, a filtered subset of LAION2B drawn from Radenovic et al., 2023. At ViT-B-16 scale, each training run requires approximately \~11,500 A100 GPUhours, which assuming a cost of \~3 USD/GPUhours on a public cloud, results in a cost of \~35,000 USD for a single training run. At ViT-L-14 scale, each training run requires approximately 40,000 A100 GPU-hours (3.5x of ViT-B-16 training cost), for a total cost of \~122,500 USD dollars per training run. Reproducing our results at ViT-L-14 scale would require at minimum training SemDeDup and baselines at several different pruning rates, requiring training 10 models for a total cost of 1.2 million USD. As such, while we agree that results on ViT-L (or ViT-H or ViT-G) would all be interesting and valuable, unfortunately, the computational cost would be prohibitively expensive and we think it is unlikely that the results would be substantively different for a ViT-L rather than a ViT-B-16.

---

> > > > > > > > ### Author Response · Authors · 2023-05-31
> > > > > > > > **Second Response to Reviewer "Liyc": Part 4**
> > > > > > > >
> > > > > > > > (vi) Experimental study
> > > > > > > >
> > > > > > > > We modified the experiment description in Section 4.1 under “CLIP Training” to reflect the number of examples seen during training by each model to make it easier to understand.
> > > > > > > >
> > > > > > > > Here is a quote from that section:
> > > > > > > >  “…For example, training on the LAION-440M dataset for 32 epochs corresponds to 14B examples seen during training, and training on 50\% of it (220M examples) for 32 epochs corresponds to 7B (14 x 0.5) examples seen…”
> > > > > > > >
> > > > > > > > And  also:
> > > > > > > >
> > > > > > > > “…In addition to fixing the batch size for all models, we also fix the number of GPUs used for training,
> > > > > > > > specifically, we use 176 A100 GUPs…”
> > > > > > > >
> > > > > > > > We have also changed  Fig. 1 and 5 and added the “Number of Samples Seen During Training” to the x-axis to make the relationship between the number of samples seen and iterations clearer.
> > > > > > > >
> > > > > > > > We summarize our experimental design after modifying the description as follows:
> > > > > > > > Fix the hyperparameters (including the batch size), the model, and the number of GPUs.
> > > > > > > > 1. For training CLIP models on the whole LAION-440M dataset, we train for 32 epochs on 440M examples, corresponding to 14B examples seen during training.
> > > > > > > > 2. For training on 50\% of the LAION-440M dataset, we train for 32 epochs on 220M examples (0.5 x 440M), corresponding to 7B (0.5 x 14B) examples seen during training.
> > > > > > > > 3. For training on p\% of the LAION-440M dataset, we train for 32 epochs on ((p\%/100) x 440M) examples, corresponding to ((p\%/100) x 14B) examples seen during training.
> > > > > > > >
> > > > > > > > (vii) Using pretrained model
> > > > > > > >
> > > > > > > > Regarding the model used for embedding the LAION-440M dataset for the results in Fig. 4 and 5 see Section 4.4 under “Training on semantically deduplicated data improves efficiency”
> > > > > > > >
> > > > > > > > “....For SemDeDup, we use embeddings from a CLIP model pretrained on LAION-440M. We show later in Section 6.2 that SemDeDup is robust to the choice of the pretrained model….”
> > > > > > > >
> > > > > > > >
> > > > > > > > To further investigate the reviewer's request, we conducted additional experiments, and found that SemDeDup still works well with a smaller and simpler embedding: we were able to achieve comparable performance with a smaller model trained on ImageNet. We do note that the CLIP-Base model is a relatively small model with around 150M parameters across both the image encoder and the text encoder, though we only use the image encoder in this work to prune the data.
> > > > > > > >
> > > > > > > >
> > > > > > > > As an additional experiment, we trained a much smaller DINO-Base model with only 85M parameters (which is 2x smaller than CLIP-Base), pre-trained on a much smaller dataset (ImageNet-1K, which is 336 times smaller than LAION). We found that pruning using these embeddings from DINO yielded very similar performance (66.9\% for CLIP embeddings vs 66.77\% for DINO embeddings). This shows that the performance of SemDeDup is relatively robust to significant changes to both the size of the training data and the size of the model used to find the embeddings. We have added this new result on the performance of SemDeDup using pruning based on DINO/ImageNet-1K to table 2.
> > > > > > > >
> > > > > > > >
> > > > > > > > These results demonstrate that you do not need to train a model on the same data in order to effectively use SemDeDup. Furthermore, we believe that SemDeDup will be practical for the vast majority of problems as pre-trained embedding models can be readily downloaded from the internet for most domains.

---

> > > > > > > > > ### Comment · Reviewer_Liyc · 2023-06-02
> > > > > > > > > **Thoughts on second response**
> > > > > > > > >
> > > > > > > > > Thanks for your updated response! Quickly:
> > > > > > > > >
> > > > > > > > > Novelty: The paper provides empirical evidence for coreset selection in the embedding space. It does that by (i) using a heuristic to compute these coresets scalably (=SemDeDup) and (ii) an experimental study at scale. This is how -- in my opinion -- the paper should be positioned. Much of the discussion in the introduction, for example, is what motivated the use of coresets in the first place.
> > > > > > > > >
> > > > > > > > > Method: It's clear that k-means + all-similarity is more effective in practice than a naive approach, and the heuristic seems to perform reasonably well. But it's a "just" heuristic; this problem has been considered before (e.g., SimHash/LSH or FAISS directly are applicable to cosine similarity). But, reiterating: "So, yes the proposed method is not very complex, but no, it's not the simplest method that comes to mind. Again, and as stated in my original review, all of this would be OK with me if the paper discussed in adequately. It does not, however."
> > > > > > > > >
> > > > > > > > > Comparisons: The comparison to Sorcher et al. (2022) in the appendix is helpful (although the argument probably falls apart when the number of clusters goes up). There is no comparison to libraries such as fastdup, which received signifcant attention recently and is a major omission. Again, if the paper were positioned as in (i), that would be ok. It's not that SemDeDup outperforms (all) prior work, it's that it provides evidence that pruning in embedding space can work well.
> > > > > > > > >
> > > > > > > > > SOTA?: Quoting: "My main gripe is that the obtained accuracy (69%) is not related to what's possible on Laion440M (e.g., OpenCLIP reports 72.77% with ViT-L/14). This point would be easy to fix, e.g., by stating/arguing for it or (preferably) by adding results with ViT-L/14." -- The authors argue that (understandably) the second fix is too costly. To me, the first one is also OK: the paper should spell out the limitations that it's unclear whether deduplicated data is also beneficial for larger models.
> > > > > > > > >
> > > > > > > > > For the study:
> > > > > > > > >
> > > > > > > > > It may help to write the "fixed values above each figure (e.g., "32 epochs").
> > > > > > > > >
> > > > > > > > > The entire random pruning results are not really insightful to me; it's even worse than simply using the full dataset (which would be visible if random pruning would be shown in Fig 5b). Results such as the ones in Tab A1 are more insightful (but, again, here the number of used examples needs to be "computed" by the reader.)
> > > > > > > > >
> > > > > > > > > Tab 2 may show the Top1/Top5 results for the base extraction models as well. This would (hopefully) make the argument clearer that using a weak extraction model is OK.
> > > > > > > > >
> > > > > > > > > Statements such as "For SemDeDup, we use embeddings from a CLIP model pretrained on LAION-440M." should be clarified. Which CLIP  model?
> > > > > > > > >
> > > > > > > > > Overall, my key concern remains the positioning of the paper.

---

> > > > > > > > > > ### Author Response · Authors · 2023-06-02
> > > > > > > > > > **Third response**
> > > > > > > > > >
> > > > > > > > > > Thank you for your prompt reply!
> > > > > > > > > >
> > > > > > > > > > Regarding the positioning of the paper, we disagree that our paper is positioned to either a) claim novelty of the method, or b) state that our method is the "simplest" - there are of course simpler (though less effective or computationally intractable) approaches available. We simply claim that the method is effective at web scale while remaining quite simple (not necessarily the "simplest"). As mentioned in our previous response, the word "novel" is never used to describe SemDeDup, nor does the word "simplest" appear anywhere in the manuscript.
> > > > > > > > > >
> > > > > > > > > > However, we certainly do not want to mislead readers, so as stated in our previous response, we are happy to change any specific lines or passages the reviewer feels are misleading or which position our manuscript incorrectly.

---

### Decision · Action_Editors · 2023-06-06

**Recommendation:** Reject

**Comment:**

Although one of the reviewers is positive about the content of the paper, the other two reviewers raised issues about novelty and solidity of the empirical assessment. Because of that, the current version of the paper cannot be considered for publication.

**Audience:**

The paper covers a topic of interest for TMLR audience, however the current form of presentation and reported empirical results only partially support the authors claims, thus making the authors' findings to lack a solid base.

**Claims And Evidence:**

The novelty of the proposed approach is not clear, since it combines ideas already presented in the relevant literature. Presentation does not address this issue properly, i.e., where is the novelty with respect to already known ideas on pruning in embedding space and scalability ?
The proposed approach is mainly heuristic and does not hinge on a principled criterion. Moreover, the provided empirical evidence does not show SOTA results in all cases, and does not cover the full spectrum of possibility, like large models and stronger baselines.
In summary, a better positioning of the contribution and more empirical evidence is needed to provide a solid support to the authors' claims.

**Resubmission Of Major Revision:**

The authors may consider submitting a major revision at a later time.